# Nucleotide inhibition of the pancreatic ATP-sensitive K$^+$ channel explored with patch-clamp fluorometry

Samuel G Usher, Frances M Ashcroft*, Michael C Puljung*

Department of Physiology, Anatomy and Genetics, University of Oxford, Oxford, United Kingdom

**Abstract** Pancreatic ATP-sensitive K$^+$ channels (K$_{ATP}$) comprise four inward rectifier subunits (Kir6.2), each associated with a sulphonylurea receptor (SUR1). ATP/ADP binding to Kir6.2 shuts K$_{ATP}$. Mg-nucleotide binding to SUR1 stimulates K$_{ATP}$. In the absence of Mg$^{2+}$, SUR1 increases the apparent affinity for nucleotide inhibition at Kir6.2 by an unknown mechanism. We simultaneously measured channel currents and nucleotide binding to Kir6.2. Fits to combined data sets suggest that K$_{ATP}$ closes with only one nucleotide molecule bound. A Kir6.2 mutation (C166S) that increases channel activity did not affect nucleotide binding, but greatly perturbed the ability of bound nucleotide to inhibit K$_{ATP}$. Mutations at position K205 in SUR1 affected both nucleotide affinity and the ability of bound nucleotide to inhibit K$_{ATP}$. This suggests a dual role for SUR1 in K$_{ATP}$ inhibition, both in directly contributing to nucleotide binding and in stabilising the nucleotide-bound closed state.

*For correspondence:
frances.ashcroft@dpag.ox.ac.uk
(FMA);
michael.puljung@dpag.ox.ac.uk
(MCP)

Competing interests: The authors declare that no competing interests exist.

## Introduction

ATP-sensitive K$^+$ channels (K$_{ATP}$) couple the metabolic state of a cell to its electrical activity (*Ashcroft and Rorsman, 2013*). In pancreatic β-cells, closure of K$_{ATP}$ in response to glucose uptake triggers insulin secretion. As such, mutations in K$_{ATP}$ that affect its response to changes in cellular metabolism cause diseases of insulin secretion, for example neonatal diabetes and persistent hyperinsulinemic hypoglycaemia of infancy (PHHI; *Quan et al., 2011*; *Ashcroft et al., 2017*). K$_{ATP}$ is composed of four inwardly rectifying K$^+$ channel subunits (Kir6.2 in pancreatic β-cells), which form the channel pore, and four modulatory sulphonylurea receptor subunits (SUR1 in β-cells; *Figure 1A*; *Aguilar-Bryan et al., 1995*; *Inagaki et al., 1995*; *Sakura et al., 1995*; *Inagaki et al., 1997*). SUR1 is a member of the ABC transporter family but lacks any transport activity (*Aguilar-Bryan et al., 1995*; *Tusnády et al., 1997*). K$_{ATP}$ responds to metabolism via adenine nucleotide binding to three distinct classes of intracellular nucleotide-binding site (one on each Kir6.2 subunit and two on each SUR1 subunit—making twelve sites in total [*Vedovato et al., 2015*]). Binding of ATP or ADP to Kir6.2 inhibits K$_{ATP}$ channel activity (*Tucker et al., 1997*; *Proks et al., 2010*), whereas binding of nucleotides to SUR1 stimulates K$_{ATP}$ (*Nichols et al., 1996*; *Tucker et al., 1997*). The stimulatory activity of nucleotides on K$_{ATP}$ depends on Mg$^{2+}$ (*Gribble et al., 1998*), whereas their inhibitory effect on Kir6.2 does not (*Tucker et al., 1997*).

In addition to nucleotide-dependent activation, SUR1 confers several other properties on Kir6.2. First, association with SUR1 increases the open probability ($P_{open}$) of Kir6.2 (*Babenko and Bryan, 2003*; *Chan, 2003*; *Fang et al., 2006*). Despite this increase in $P_{open}$, SUR1 also paradoxically increases the apparent affinity for nucleotide inhibition at Kir6.2 by an unknown mechanism (*Tucker et al., 1997*). SUR1 is also responsible for high-affinity inhibition of K$_{ATP}$ by antidiabetic sulphonylureas and glinides as well as activation by K$_{ATP}$-specific K$^+$ channel openers (*Tucker et al.,*

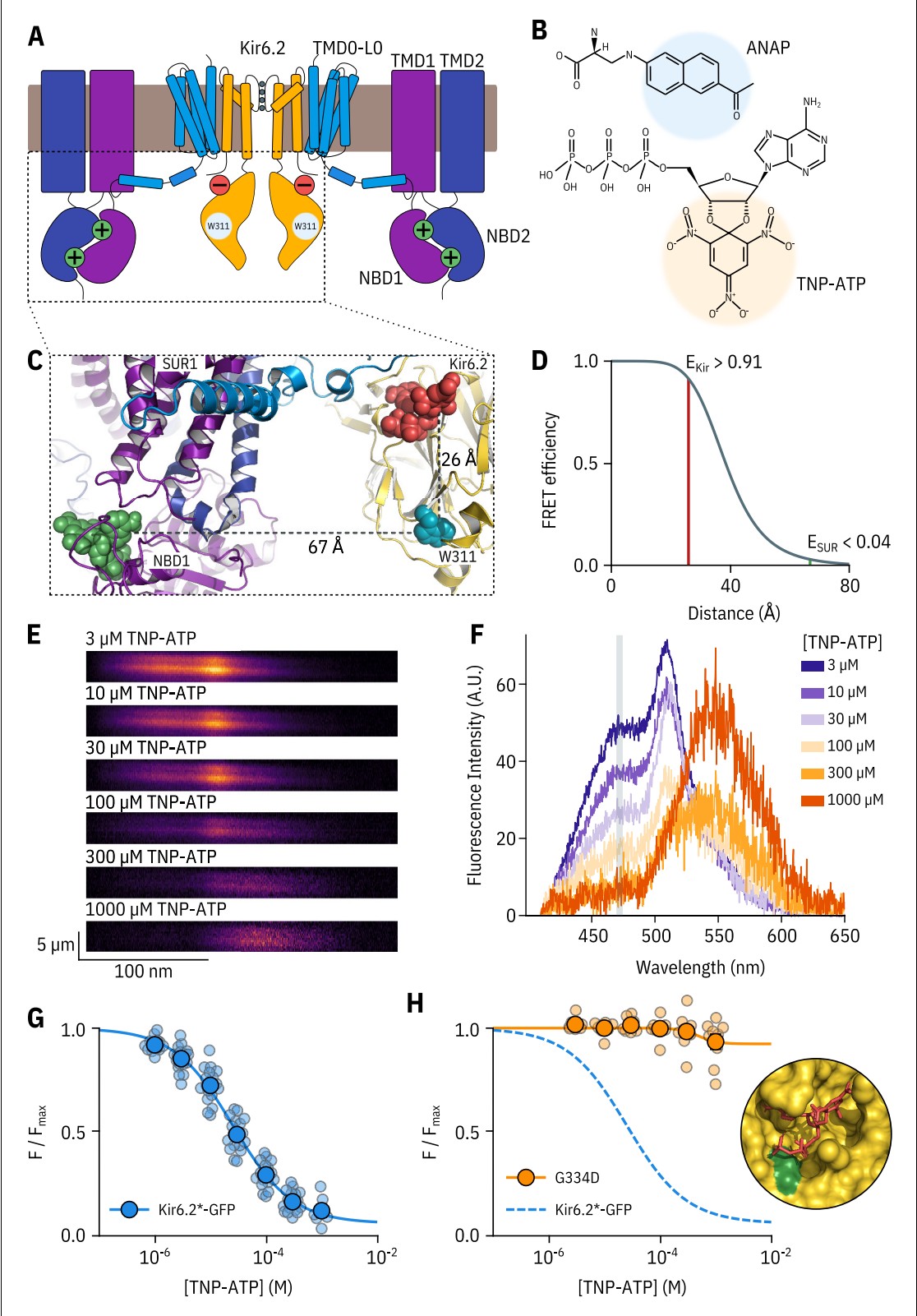

**Figure 1.** A FRET assay to measure nucleotide binding to Kir6.2. (**A**) Cartoon illustrating the topology of $K_{ATP}$. Two (of four) Kir6.2 and two (of four) SUR1 subunits are shown for clarity. The inhibitory nucleotide-binding site on Kir6.2 is shown in red; the stimulatory nucleotide-binding sites on SUR1 are shown in green. The three transmembrane domains of SUR1 are designated TMD0, TMD1, and TMD2. The loop connecting TMD0 to TMD1 is designated L0. The nucleotide binding domains of SUR1 are labelled NBD1 and NBD2. (**B**) Chemical structures of ANAP and TNP-ATP. The fluorescent

*Figure 1 continued on next page*

*Figure 1 continued*

moieties are highlighted. (C) Side view of the structure of the cytosolic domains of Kir6.2 (PDB accession #6BAA) and one SUR1 subunit (PDB accession #6PZI). TNP-ATP (red, from PDB accession #5XW6) was docked into the nucleotide-binding site of Kir6.2 and positioned in NBS1 of SUR1 (green, from PDB accession #3AR7) by alignment as described in Materials and methods. Distances from the centre of mass of the six-membered ring of the native tryptophan at position 311 in Kir6.2 to the centre of mass of the trinitrophenyl moieties of the TNP-ATPs are displayed in Å. (D) Theoretical FRET efficiency between ANAP and TNP-ATP as a function of distance, calculated from the Förster equation. The distances and corresponding FRET efficiencies between ANAP at position 311 and TNP-ATP bound to Kir6.2 ($E_{Kir}$) and SUR1 ($E_{SUR}$) are indicated. Our calculated $R_0$ (the distance at which FRET efficiency is half maximal) for ANAP and TNP-ATP is 38.4 Å. (E) Spectral images acquired from an unroofed membrane expressing Kir6.2*-GFP + SUR1 and exposed to increasing concentrations of TNP-ATP. The y-dimension in each image represents distance. The x-dimension represents wavelength. (F) Line-averaged, background-subtracted spectra from E displayed with increasing concentrations of TNP-ATP coloured from purple to orange. The three fluorophores have distinct peaks: ANAP at 472 nm, GFP at 508 nm, and TNP-ATP at 561 nm. The shaded rectangle indicates the wavelength range used to measure ANAP intensity. (G) Concentration-response relationship for binding of TNP-ATP to Kir6.2*-GFP + SUR1 in unroofed membranes. Data were plotted as $F/F_{max}$, where $F_{max}$ is the fluorescence intensity of ANAP in the absence of nucleotide. The smooth curve is a descriptive Hill fit. $EC_{50} = 25.6\,\mu M$, $h = 0.82$, $E_{max} = 0.93$, n = 18. (H) Concentration-response relationship for binding of TNP-ATP to Kir6.2*,G334D-GFP + SUR1 in unroofed membranes. The dashed blue curve is the fit from G. The orange curve is a descriptive Hill fit to the G334D data. $EC_{50} = 493\,\mu M$, $h = 2.63$, $E_{max} = 0.08$, n = 9. The inset shows the location of G334D (green) in relation to the inhibitory ATP binding site on Kir6.2 (PDB accession #6BAA). TNP-ATP (PDB accession #5XW6) shown in red sticks.

The online version of this article includes the following figure supplement(s) for figure 1:

**Figure supplement 1.** ANAP labelling is specific and only full-length Kir6.2 is expressed at the cell membrane.
**Figure supplement 2.** Kir6.2*-GFP is functionally similar to Kir6.2-GFP.

*1997*). Finally, SUR1 and Kir6.2 must co-assemble to ensure mutual exit from the endoplasmic reticulum and correct trafficking to the plasma membrane (*Zerangue et al., 1999*).

To date, the primary means of studying nucleotide-dependent effects on $K_{ATP}$ channel function has been with electrophysiological approaches, which measure the summed activity of all three classes of binding site acting in concert. Thus, it can be difficult to separate the contributions of each class of site to the opening and closing of the channel pore and to properly distinguish between changes in nucleotide binding and channel gating. To overcome these limitations, we have applied a novel approach to directly measure nucleotide binding to each individual class of site in $K_{ATP}$ (*Puljung et al., 2019*). This method utilizes Förster resonance energy transfer (FRET) between channels labelled with the fluorescent unnatural amino acid 3-(6-acetylnaphthalen-2-ylamino)−2-aminopropanoic acid (ANAP) and fluorescent trinitrophenyl (TNP) analogues of adenine nucleotides (*Figure 1B*). As we show here, this method is readily combined with patch-clamp electrophysiology so that nucleotide binding and regulation of current can be measured simultaneously. This has enabled us to quantitatively assess nucleotide binding to Kir6.2 and explore how this is coupled to channel inhibition in both wild-type $K_{ATP}$ and $K_{ATP}$ carrying mutations that impair ATP inhibition.

## Results

### Measuring nucleotide binding to Kir6.2

We previously used this FRET-based binding assay to measure nucleotide binding to the second nucleotide-binding site of SUR1 (*Puljung et al., 2019*). To measure binding to Kir6.2 in the complete $K_{ATP}$ complex (four full-length Kir6.2 subunits co-expressed with four full-length SUR1 subunits), we replaced a tryptophan at position 311 (W311) of Kir6.2 that is 26 Å from the location of the inhibitory nucleotide-binding site with ANAP (*Figure 1C*) such that each subunit is labelled with one ANAP molecule. We designate this construct Kir6.2*. Based on the theoretical FRET efficiency calculated from the Förster equation and available cryo-EM structures (*Martin et al., 2017*; *Martin et al., 2019*), we expect 91% FRET efficiency between ANAP at position 311 and a TNP-ATP molecule bound to Kir6.2, and only 4% FRET efficiency to TNP-ATP bound to the closest nucleotide-binding site on SUR1 (nucleotide binding site 1; *Figure 1C; Figure 1D*). We also expect very little FRET between ANAP at position 311 and TNP-ATP bound to neighbouring Kir6.2 subunits (see Materials and methods for more details on our calculations).

ANAP incorporation into Kir6.2 was achieved as described previously (*Chatterjee et al., 2013*; *Zagotta et al., 2016*; *Puljung et al., 2019*). Briefly, HEK-293T cells were co-transfected with a plasmid encoding a Kir6.2 construct with a C-terminal GFP tag and an amber stop codon (TAG)

replacing the codon corresponding to amino acid position 311 (W311[TAG]-GFP) and a plasmid encoding an ANAP-specific tRNA/tRNA synthetase pair (pANAP). We also included a dominant negative eukaryotic ribosomal release factor (eRF-E55D) in our transfections, which has been shown to increase the yield of full-length, ANAP-labelled protein (*Schmied et al., 2014*; *Puljung et al., 2019*). When cultured in the presence of ANAP, full length, fully ANAP-labelled Kir6.2 protein was produced and successfully trafficked to the membrane in the presence of SUR1 (*Figure 1—figure supplement 1*; see Materials and methods). We used GFP-tagged Kir6.2 constructs throughout this study unless otherwise indicated, to help identify cells or membranes expressing $K_{ATP}$.

Currents were measured from patches from cells expressing $K_{ATP}$ that were excised in the absence of $Mg^{2+}$. Under such conditions, nucleotides can bind to both sites on SUR1, but no activation occurs, allowing inhibitory currents to be measured in isolation (*Gribble et al., 1998*; *Ueda et al., 1999*; *Puljung et al., 2019*). Kir6.2*-GFP + SUR1 exhibited nearly identical sensitivity to ATP inhibition as Kir6.2-GFP + SUR1 (*Figure 1—figure supplement 2A*), indicating that replacement of W311 with ANAP did not affect inhibition of $K_{ATP}$. Whereas both constructs were inhibited by TNP-ATP with a higher apparent affinity relative to ATP, incorporation of ANAP resulted in channels with a slightly lower TNP-ATP sensitivity relative to wild-type ($IC_{50}$ of 6.2 μM compared to an $IC_{50}$ of 1.2 μM, *Figure 1—figure supplement 2B,C*).

Kir6.2-GFP has been demonstrated to traffic to the plasma membrane in the absence of SUR1 and form functional channels (*John et al., 1998*; *Makhina and Nichols, 1998*). In a luminescence-based, surface-expression assay, we did not detect HA-tagged Kir6.2*-GFP at the plasma membrane in the absence of SUR1 (*Figure 1—figure supplement 1E*). To verify that the currents measured in our experiments in which Kir6.2*-GFP was co-transfected with SUR1 were the result of Kir6.2*-GFP + SUR1 and not Kir6.2*-GFP alone, we measured the sensitivity of currents to inhibition by the sulphonylurea tolbutamide, a property conferred by the SUR1 subunit. Whereas currents from unlabelled wild-type Kir6.2-GFP expressed in the absence of SUR1 were not affected by 100 μM tolbutamide, both wild-type Kir6.2-GFP and Kir6.2*-GFP currents were inhibited to a similar extent when expressed with SUR1 (46.5% ±0.04% and 57.7% ±0.02%, respectively; *Figure 1—figure supplement 2D*). The extent of inhibition was similar to previous measurements of tolbutamide inhibition (*Tucker et al., 1997*), confirming that Kir6.2*-GFP was co-assembled with SUR1 at the plasma membrane.

To measure nucleotide binding, cells transfected with Kir6.2*-GFP + SUR1 were briefly sonicated, leaving behind unroofed plasma membrane fragments (*Heuser, 2000*; *Zagotta et al., 2016*; *Puljung et al., 2019*) containing ANAP-labelled $K_{ATP}$ channels with the intracellular nucleotide-binding sites exposed to the bath solution. The sample was excited with a 385 nm LED and emitted fluorescence from the membrane fragments was passed through a spectrometer, allowing us to separate ANAP, GFP, and TNP-ATP fluorescence by peak wavelength (*Figure 1E,F*). As expected from FRET, increasing the concentration of TNP-ATP caused a decrement in the ANAP peak at 472 nm and a concomitant increase in the TNP-ATP peak at 561 nm (*Figure 1F*). We used the quenching of the ANAP peak as a direct measure of TNP-ATP binding as this signal was specific to $K_{ATP}$. In contrast, the peak TNP-ATP fluorescence may include contributions from both specific and non-specific nucleotide binding, as well as direct excitation of TNP-ATP in solution by the 385 nm excitation light. Due to the sharp cut-off of the GFP emission spectrum at shorter wavelengths, our measurements of peak ANAP fluorescence were unaffected by the presence of the GFP tag on Kir6.2.

We fit concentration-response data for ANAP quenching by TNP-ATP with the Hill equation, to produce estimates of apparent affinity ($EC_{50}$, the half maximal effective concentration) and $E_{max}$ (ANAP quenching at saturating concentrations of TNP-ATP; *Figure 1G*). $E_{max}$ was 93%, in good agreement with the 91% predicted by the Förster equation and theoretical distance measurements (*Figure 1D*), suggesting that we were able to measure binding directly to the inhibitory site at Kir6.2. To confirm this, we introduced a well-studied neonatal diabetes mutation (G334D) into the Kir6.2 binding site, which drastically reduces the sensitivity of the channel to inhibition by nucleotides (*Drain et al., 1998*; *Masia et al., 2007*; *Proks et al., 2010*). Based on the cryo-electron microscopy structures of $K_{ATP}$, this mutation is expected to interfere with nucleotide binding directly (*Figure 1H* inset, *Martin et al., 2017*). The resulting construct Kir6.2*,G334D-GFP + SUR1 displayed drastically reduced ANAP quenching over the range of TNP-ATP concentrations tested. We therefore conclude that our binding measurements were specific for the inhibitory nucleotide-binding site

on Kir6.2. This observation is consistent with the interpretation that the G334D mutation causes neonatal diabetes by preventing nucleotide binding.

## Measuring current inhibition and nucleotide binding simultaneously

The apparent affinity of Kir6.2*-GFP + SUR1 for TNP-ATP in unroofed membranes was 25.6 μM (*Figure 1G* and *Table 1*). This value is higher than the apparent affinity for nucleotide inhibition (6.2 μM) measured using patch-clamp (*Figure 1—figure supplement 2C*). However, both binding and current measurement are a function of the intrinsic binding affinity, the channel $P_{open}$, and the ability of agonist, once bound, to close the channel. Furthermore, the functional state of $K_{ATP}$ in unroofed membranes is unclear. This is a particular problem with $K_{ATP}$ channels, which run down due to slow dissociation of phosphatidylinositol 4,5-bisphosphate ($PIP_2$), reducing the $P_{open}$ over time even in the absence of nucleotides (*Proks et al., 2016*).

As measuring either nucleotide binding or ionic currents in isolation only offers limited mechanistic insight into inhibition of $K_{ATP}$, we turned to patch-clamp fluorometry (PCF, *Zheng and Zagotta, 2003*). Using PCF, we can measure TNP-ATP binding to Kir6.2 and channel activity simultaneously (*Figure 2*), providing us with direct access to the relationship between nucleotide binding and channel function. We simultaneously measured fluorescence emission spectra and ionic currents for Kir6.2*-GFP + SUR1 in inside-out, excised membrane patches. The apparent negative fluorescence intensities at high TNP-ATP concentrations are due to imperfect background subtraction, and do not affect our measurements of ANAP intensities (see Materials and methods). As before, all measurements were performed in the presence of $Mg^{2+}$ chelators, such that nucleotide inhibition could be measured in the absence of activation (*Tucker et al., 1997*; *Gribble et al., 1998*). Strikingly, current inhibition occurred at a lower range of concentrations compared to nucleotide binding (*Figure 2C,D*). The apparent $IC_{50}$ for inhibition calculated from Hill fits was an order of magnitude lower than the $EC_{50}$ for binding measured in the same patches (*Figure 2C*, *Table 2*). We considered several different gating models to explain this observation. In each model, we assumed the channel pore was able to open and close in the absence of ligand with an equilibrium constant $L$, where $P_{open} = L/(L+1)$ and $L > 0$, reflecting the ability of $K_{ATP}$ to open and close in the absence of nucleotides. This excludes the possibility of induced-fit models which would not predict unliganded channel closings. Induced-fit models also cannot account for separation between the binding and gating curves which we observe in *Figure 2C,D* (*Changeux and Edelstein, 2011*). Each model also had parameters representing the intrinsic binding affinity to the closed state ($K_A$, where $K_A > 0$) and the factor by which nucleotide binding favours channel closure ($D$, where $D < 1$).

Our simultaneous binding and current measurements were well fit with a Monod-Wyman-Changeux (MWC)-type model (*Figure 2D,E*; *Monod et al., 1965*) which has been previously

**Table 1.** Hill fit parameters from unroofed membranes.

$EC_{50}$ values and their standard errors are reported as $\log_{10} M$. $EC_{50}$ values are also provided as μM in parentheses.

| Fluorescence quenching | Construct | Term | Estimate | Standard error |
|---|---|---|---|---|
| *TNP-ATP* | Kir6.2*-GFP+SUR1 | $EC_{50}$ | −4.59 (25.7) | 0.05 |
| | n = 18 | $h$ | 0.82 | 0.05 |
| | | $E_{max}$ | 0.93 | 0.03 |
| | Kir6.2*,G334D-GFP+SUR1 | $EC_{50}$ | −3.31 (490) | 2.23 |
| | n = 9 | $h$ | 2.63 | 17.70 |
| | | $E_{max}$ | 0.08 | 0.26 |
| | Kir6.2*,C166S-GFP+SUR1 | $EC_{50}$ | −4.50 (31.6) | 0.05 |
| | n = 12 | $h$ | 0.92 | 0.08 |
| | | $E_{max}$ | 0.87 | 0.03 |
| | Kir6.2*-GFP | $EC_{50}$ | −4.42 (38.0) | 0.05 |
| | n = 14 | $h$ | 0.83 | 0.05 |
| | | $E_{max}$ | 0.92 | 0.03 |

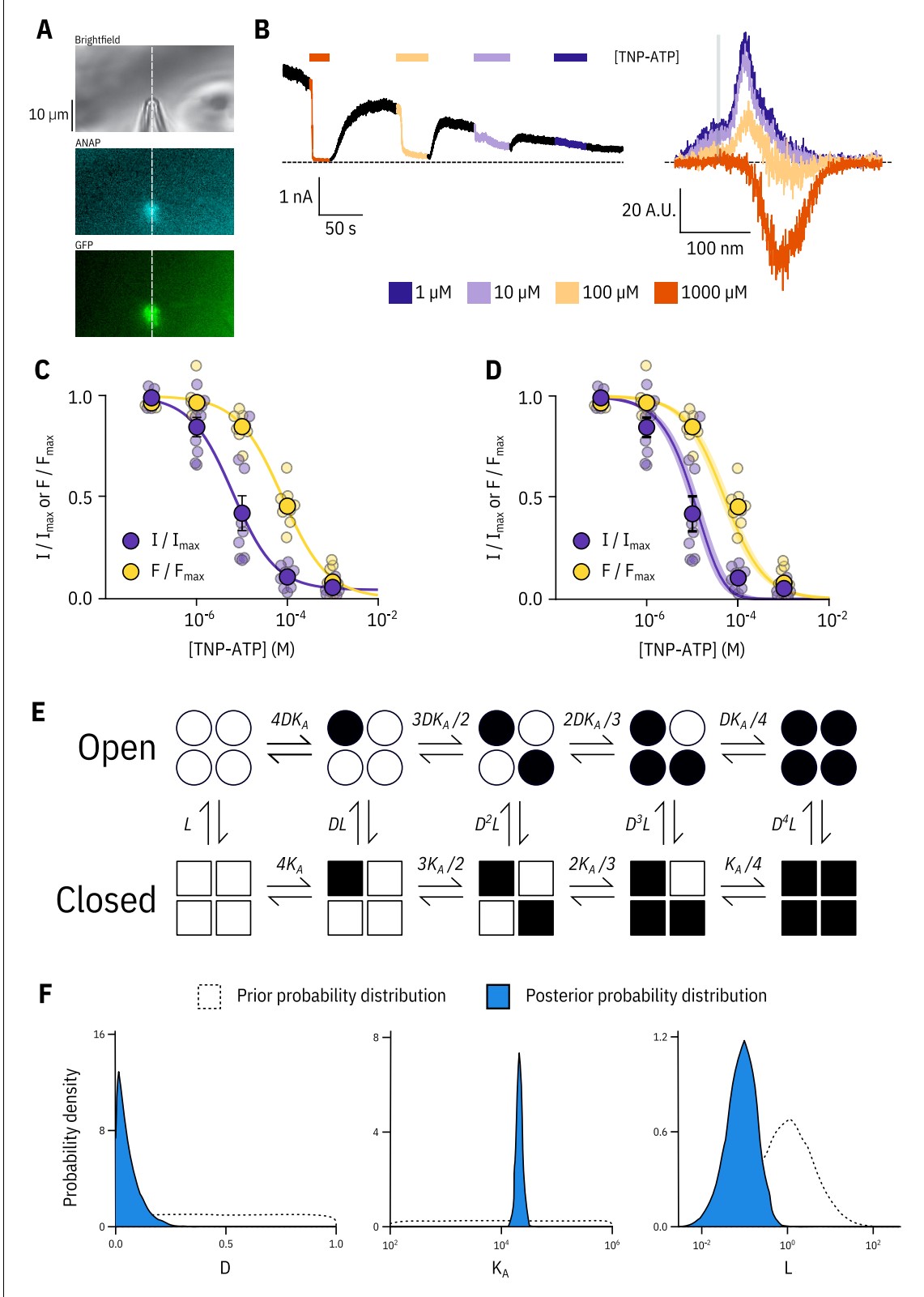

**Figure 2.** Simultaneous measurements of nucleotide binding and channel current. (**A**) Brightfield and fluorescence images of a patch pipette and excised, inside out patch expressing Kir6.2*-GFP + SUR1, with the location of the centre of the spectrometer slit overlaid as a white, vertical line. (**B**) Current (left) and spectra (right) acquired from the same excised, inside-out patch exposed to TNP-ATP and coloured according to concentration. (**C**) Concentration-response (n = 9) for TNP-ATP inhibition of Kir6.2*-GFP + SUR1 currents ($I/I_{max}$) and for quenching of ANAP fluorescence ($F/F_{max}$). Both

*Figure 2 continued on next page*

*Figure 2 continued*

current inhibition and fluorescence quenching were fit to the Hill equation. Current inhibition: $IC_{50} = 6.23\,\mu M$, $h = 0.92$, $I_{max} = 0.96$, fluorescence quenching: $EC_{50} = 77.7\,\mu M$, $h = 0.87$, $E_{max} = 1.00$. (D) The same data as in C fit to an MWC-type model. Solid curves represent the median fit; shaded areas represent the 95% quantile interval. Values for the fits are reported in the text and in *Table 3*. (E) MWC-type model for inhibition of K$_{ATP}$ by nucleotides. Open subunits are shown as circles; closed are shown as squares. Nucleotide-bound subunits are represented by filled symbols. *L*, *D*, and $K_A$ are defined in the text. (F) Posterior probability distributions for the MWC-type model generated by MCMC fits to the data in C overlaid on the prior probability distribution (dashed line) for each parameter.

The online version of this article includes the following figure supplement(s) for figure 2:

**Figure supplement 1.** Fixing *L* does not affect estimates of *D* and $K_A$.
**Figure supplement 2.** Model selection.
**Figure supplement 3.** Bleaching correction for PCF experiments.

proposed to explain K$_{ATP}$ channel inhibition (*Enkvetchakul and Nichols, 2003*; *Craig et al., 2008*; *Vedovato et al., 2015*). In our MWC-type model, each ligand binding event ($K_A$) is independent and each bound ligand favours the closed state by the same factor (*D*). Simultaneous measurement of binding (fluorescence) and gating (current) allowed us to obtain well constrained fits to our model. To obtain free parameter (*L*, $K_A$, *D*) estimates and verify that each parameter was well and uniquely determined, we employed a Bayesian Markov chain Monte Carlo (MCMC) method previously employed by *Hines et al. (2014)*. Using this approach, we constructed posterior probability distributions for the free parameters of our MWC-type model (*Figure 2F*, *Table 3*). Based on these distributions, we estimated $K_A = 2.1 \times 10^4\,M^{-1}$ ($K_D = 47.9\,\mu M$), $L = 0.09$ ($P_{open} = 0.08$), and $D = 0.04$. The very low *D* value indicates that nucleotide binding was tightly coupled to channel closure; that is nucleotides have a very strong preference for the closed state of the channel. The low value for *D* also explains why the channels were nearly completely inhibited at TNP-ATP concentrations at which not all the binding sites were occupied, as well as the degree to which channel inhibition is complete at saturating concentrations of TNP-ATP. Our estimate of *L* was quite low and broadly distributed. We repeated our fits with *L* fixed to a value consistent with previous single channel measurements (0.8, $P_{open} = 0.45$; *John et al., 1998*; *Enkvetchakul et al., 2000*; *Ribalet et al., 2006*). This had only a very small effect on our estimates of *D* and $K_A$ (*Figure 2—figure supplement 1*). The broad distribution of *L* in our fit may represent current rundown which occurs during our patch-clamp recordings and is expected to affect the open-closed equilibrium. Cross-correlation plots (in parameter space) of the values derived from our fits produced well bounded ellipsoids, indicating that our parameters were uniquely determined (*Figure 2—figure supplement 1A*).

In addition to the full MWC-type model we considered alternate models (*Figure 2—figure supplement 2*). These included a model in which only the first binding event influences the open-closed equilibrium of the channel (single-binding model; *Figure 2—figure supplement 2B*, *Table 3*), and an MWC-style model with an additional parameter *C* to allow for direct negative cooperativity between binding sites (negative cooperativity model; *Figure 2—figure supplement 2C*, *Table 3*). The single-binding model yielded very similar parameter estimates to our full MWC-type model (*Figure 2—figure supplement 2D*, *Table 3*). This is a consequence of *D* being so low that even in the MWC-type model most channels are closed when only a single nucleotide is bound. The cooperative model improved our fits, but not enough to justify the inclusion of an additional free parameter (see Discussion).

## Kir6.2-C166S affects the ability of bound nucleotides to close K$_{ATP}$

To provide a rigorous test as to whether our experimental system was capable of separating nucleotide binding from subsequent channel gating, we introduced a mutation (Kir6.2-C166S) which increases $P_{open}$ of K$_{ATP}$ and decreases sensitivity of the channel to inhibition by nucleotides (*Trapp et al., 1998*). C166 is located near the bundle-crossing gate of Kir6.2 (*Figure 3A*). Other mutations at this site cause neonatal diabetes (*Flanagan et al., 2006*; *Gloyn et al., 2006*).

In unroofed membranes, Kir6.2*,C166S-GFP + SUR1 bound TNP-ATP with an $EC_{50}$ very similar to that of Kir6.2*-GFP + SUR1 (*Figure 3B*, 32.0 μM and 25.6 μM, respectively), which suggests only a small change in apparent nucleotide affinity. This is an unexpected finding, as one might expect that an increase in $P_{open}$ would allosterically cause a decrease in the apparent affinity for inhibitory

**Table 2.** Hill fit parameters from excised patches.

$EC_{50}$ and $IC_{50}$ values and their standard errors are reported as $\log_{10} M$. $EC_{50}$ and $IC_{50}$ values are also provided as μM in parentheses.

| Current inhibition | Construct | Term | Estimate | Standard error |
|---|---|---|---|---|
| ATP | Kir6.2-GFP + SUR1 | $IC_{50}$ | −4.20 (63.1) | 0.07 |
| | n = 3 | $h$ | 1.28 | 0.21 |
| | | $I_{max}$ | 0.99 | 0.06 |
| | Kir6.2-GFP | $IC_{50}$ | −3.31 (490) | 0.05 |
| | n = 2 | $h$ | 1.15 | 0.12 |
| | | $I_{max}$ | 0.93 | 0.03 |
| | Kir6.2*-GFP + SUR1 | $IC_{50}$ | −4.10 (79.4) | 0.06 |
| | n = 4 | $h$ | 1.42 | 0.21 |
| | | $I_{max}$ | 1.00 | 0.05 |
| TNP-ATP | Kir6.2-GFP + SUR1 | $IC_{50}$ | −5.93 (1.17) | 0.04 |
| | n = 7 | $h$ | 1.14 | 0.11 |
| | | $I_{max}$ | 0.97 | 0.02 |
| | Kir6.2-GFP | $IC_{50}$ | −3.56 (275) | 0.64 |
| | n = 3 | h | 1.09 | 0.85 |
| | | $I_{max}$ | 1.00 | 0.53 |
| | Kir6.2*-GFP + SUR1 | $IC_{50}$ | −5.21 (6.17) | 0.10 |
| | n = 9 | $h$ | 0.92 | 0.18 |
| | | $I_{max}$ | 0.96 | 0.05 |
| | Kir6.2*,C166S-GFP + SUR1 | $IC_{50}$ | −3.11 (776) | 0.23 |
| | n = 6 | $h$ | 1.35 | 1.16 |
| | | $I_{max}$ | 0.55 | 0.11 |
| | Kir6.2*-GFP + SUR-K205A | $IC_{50}$ | −3.78 (166) | 0.45 |
| | n = 9 | $h$ | 0.75 | 0.30 |
| | | $I_{max}$ | 1.00 | 0.29 |
| | Kir6.2*-GFP + SUR-K205E | $IC_{50}$ | −3.20 (631) | 2.15 |
| | n = 9 | $h$ | 0.79 | 0.84 |
| | | $I_{max}$ | 1.00 | 1.77 |
| **Fluorescence Quenching** | | | | |
| TNP-ATP | Kir6.2*-GFP + SUR1 | $EC_{50}$ | −4.11 (77.6) | 0.09 |
| | n = 9 | $h$ | 0.87 | 0.11 |
| | | $E_{max}$ | 1.00 | 0.06 |
| | Kir6.2*,C166S-GFP + SUR1 | $EC_{50}$ | −4.17 (67.6) | 0.23 |
| | n = 6 | $h$ | 0.84 | 0.27 |
| | | $E_{max}$ | 1.00 | 0.14 |
| | Kir6.2*-GFP + SUR-K205A | $EC_{50}$ | −3.69 (204) | 0.42 |
| | n = 9 | $h$ | 0.73 | 0.25 |
| | | $E_{max}$ | 1.00 | 0.27 |
| | Kir6.2*-GFP + SUR-K205E | $EC_{50}$ | −3.37 (427) | 1.10 |
| | n = 9 | $h$ | 0.74 | 0.47 |
| | | $E_{max}$ | 1.00 | 0.79 |

**Table 3.** Fitted parameters for the MWC-type models.
$L$, $K_A$ and their associated quantiles are reported as $\log_{10}$ values.

**Full MWC**

| Construct | Term | Estimate | 2.5% Quantile | 97.5% Quantile |
|---|---|---|---|---|
| Kir6.2*-GFP + SUR1 | $L$ | −1.05 | −1.85 | −0.45 |
| n = 9 | $D$ | 0.04 | 0.00 | 0.19 |
| | $K_A$ | 4.32 | 4.21 | 4.44 |
| Kir6.2*,C166S-GFP + SUR1 | $L$ | 0.29 | −1.04 | 1.41 |
| n = 6 | $D$ | 0.84 | 0.52 | 0.95 |
| | $K_A$ | 4.18 | 3.93 | 4.47 |
| Kir6.2*-GFP + SUR-K205A | $L$ | −0.37 | −1.34 | 0.41 |
| n = 9 | $D$ | 0.55 | 0.39 | 0.65 |
| | $K_A$ | 3.76 | 3.59 | 3.95 |
| Kir6.2*-GFP + SUR-K205E | $L$ | −0.18 | −1.25 | 0.70 |
| n = 9 | $D$ | 0.62 | 0.42 | 0.74 |
| | $K_A$ | 3.40 | 3.21 | 3.62 |

Single-site

| Construct | Term | Estimate | 2.5% Quantile | 97.5% Quantile |
|---|---|---|---|---|
| Kir6.2*-GFP + SUR1 | $L$ | −1.06 | −1.84 | −0.47 |
| n = 9 | $D$ | 0.05 | 0.01 | 0.10 |
| | $K_A$ | 4.33 | 4.22 | 4.44 |
| Kir6.2*,C166S-GFP + SUR1 | $L$ | 0.09 | −1.15 | 1.05 |
| n = 6 | $D$ | 0.70 | 0.29 | 0.91 |
| | $K_A$ | 4.15 | 3.88 | 4.43 |
| Kir6.2*-GFP + SUR-K205A | $L$ | −0.25 | −1.30 | 0.53 |
| n = 9 | $D$ | 0.18 | 0.06 | 0.32 |
| | $K_A$ | 3.62 | 3.45 | 3.83 |
| Kir6.2*-GFP + SUR-K205E | $L$ | −0.19 | −1.19 | 0.52 |
| n = 9 | $D$ | 0.30 | 0.13 | 0.47 |
| | $K_A$ | 3.31 | 3.13 | 3.50 |

Negative cooperativity

| Construct | Term | Estimate | 2.5% Quantile | 97.5% Quantile |
|---|---|---|---|---|
| Kir6.2*-GFP + SUR1 | $L$ | −0.42 | −1.38 | 0.48 |
| n = 9 | $D$ | 0.15 | 0.02 | 0.29 |
| | $K_A$ | 4.82 | 4.54 | 5.29 |
| | $C$ | 0.17 | 0.06 | 0.36 |
| Kir6.2*,C166S-GFP + SUR1 | $L$ | 0.32 | −0.96 | 1.47 |
| n = 6 | $D$ | 0.83 | 0.50 | 0.94 |
| | $K_A$ | 4.43 | 4.04 | 5.14 |
| | $C$ | 0.52 | 0.09 | 0.97 |
| Kir6.2*-GFP + SUR-K205A | $L$ | −0.16 | −1.18 | 0.64 |
| n = 9 | $D$ | 0.52 | 0.32 | 0.64 |
| | $K_A$ | 4.10 | 3.73 | 4.68 |
| | $C$ | 0.35 | 0.10 | 0.91 |
| Kir6.2*-GFP + SUR-K205E | $L$ | −1.11 | 0.99 | |
| n = 9 | $D$ | 0.58 | 0.32 | 0.73 |
| | $K_A$ | 3.71 | 3.34 | 4.41 |
| | $C$ | 0.45 | 0.10 | 0.96 |

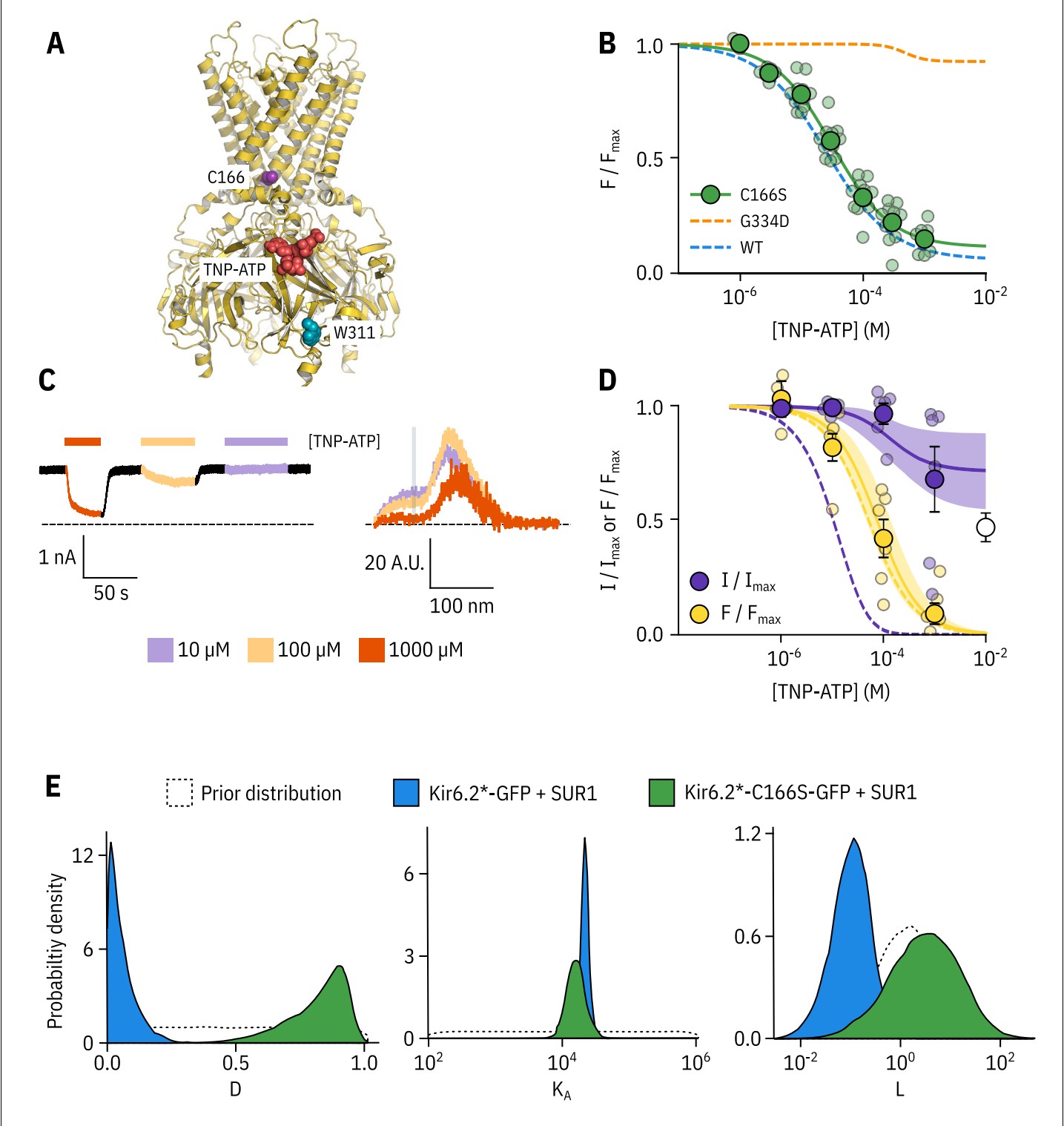

**Figure 3.** Kir6.2-C166S disrupts current inhibition, not nucleotide binding. (**A**) Cartoon (from PDB accession #6BAA) showing the location of Kir6.2-C166 (purple) relative to the inhibitory nucleotide binding site (TNP-ATP from PDB accession #5XW6 shown in red). W311 is shown as blue spheres. (**B**) Concentration dependence of TNP-ATP binding to unroofed membrane fragments expressing Kir6.2*,C166S-GFP + SUR1 shown in green, expressed as quenching of ANAP fluorescence. The Hill fits shown previously for Kir6.2*-GFP + SUR1 and Kir6.2*,G334D-GFP + SUR1 are shown in blue and orange dashed curves, respectively. Kir6.2*,C166S-GFP + SUR1: $EC_{50} = 32.0\,\mu M$, $h = 0.92$, $E_{max} = 0.96$, n = 12. (**C**) Representative current and fluorescence traces recorded simultaneously from an excised patch expressing Kir6.2*,C166S-GFP + SUR1. Exposure to different concentrations of TNP-ATP are shown by colour. (**D**) Concentration-response (n = 6) for TNP-ATP inhibition of Kir6.2*,C166S-GFP + SUR1 currents ($I/I_{max}$) and for quenching of

*Figure 3 continued on next page*

Figure 3 continued

ANAP fluorescence ($F/F_{max}$). Data were fit with the MWC-type model. Solid curves represent the median fits and shaded areas indicate the 95% quantile intervals. Dashed curves represent the previous median fits of the MWC-type model to the Kir6.2*-GFP + SUR1 data from *Figure 2D*. Parameter estimates are reported in *Table 3*. The open data point represents current inhibition by $10 \, \text{mM}$ ATP and was not included in the model fitting. (E) Posterior probability distributions for the full MWC-type model fit to Kir6.2*,C166S-GFP + SUR1 or Kir6.2*-GFP + SUR1 (data from *Figure 2F*) overlaid on the prior probability distribution.

The online version of this article includes the following figure supplement(s) for figure 3:

**Figure supplement 1.** AN MWC-type model predicts a nucleotide-insensitive current plateau for Kir6.2-C166S.

**Figure supplement 2.** Fixing $L$ does not affect the other two parameters.

nucleotide binding. To resolve this conflict, we again turned to PCF (*Figure 3C,D*). Rundown was much slower for Kir6.2*,C166S-GFP + SUR1, which may reflect the increased $P_{open}$ of this construct. Measuring current inhibition in combination with nucleotide binding confirmed that whereas the apparent nucleotide binding affinity was unchanged by the C166S mutation, current inhibition occurred at much higher concentrations compared to binding and was incomplete (*Figure 3D*). How can we explain this paradox? Fits of the data with our MWC-type model (*Figure 3D,E*) suggest that, in addition to the expected effect on $L$, the C166S mutation profoundly affects the ability of bound ligand to stabilise the closed state of the channel ($D$) without affecting $K_A$ (*Figure 3E*, *Table 3*). We propose that, in addition to increasing the $P_{open}$ of the channel, C166 is also important in the transduction pathway from the inhibitory nucleotide binding site on Kir6.2 to the channel gate.

Our MWC-type model predicts a nucleotide-insensitive current plateau at high concentrations, with the height of the plateau at saturating nucleotide concentrations given by $\frac{L \cdot D^4}{1 + L \cdot D^4}$. For example, when $L \cdot D^4 = 0.05$ we see a current plateau of just under 5%, and as $L \cdot D^4$ increases (as we see with Kir6.2*,C166S-GFP + SUR1) so does the plateau (*Figure 3—figure supplement 1C,D*). We attempted to confirm the existence of a nucleotide-insentitive current plateau for Kir6.2*,C166S-GFP + SUR1 at high concentrations of TNP-ATP, but were unable to test inhibition at concentrations of over 1 mM as our stocks of TNP-ATP were prepared from triethylammonium salts. Triethylamine concentrations of over 1 mM inhibited $K_{ATP}$ and influenced our results (*Figure 3—figure supplement 1A,B*). However, we saw only partial inhibition of Kir6.2*,C166S-GFP + SUR1 by 10 mM ATP, which supports the existence of a nucleotide-insensitive current fraction at high concentrations. This observation has been previously reported for mutations at Kir6.2-C166 in some constructs (*Trapp et al., 1998*; *Ribalet et al., 2006*) but not others (*Enkvetchakul et al., 2001*).

## Exploring the effect of SUR1 on nucleotide inhibition of K_ATP

SUR1 plays a complex role in the regulation of Kir6.2. It increases the $P_{open}$ of the channel and allows for the activation of the channel by Mg-nucleotides (*Nichols et al., 1996*; *Tucker et al., 1997*; *Babenko and Bryan, 2003*; *Chan, 2003*; *Fang et al., 2006*). However, it also increases the sensitivity of Kir6.2 to nucleotide inhibition (*Babenko and Bryan, 2003*; *Chan, 2003*; *Fang et al., 2006*). To understand the effect of SUR1 on nucleotide inhibition of K_ATP, we expressed Kir6.2*-GFP in the absence of SUR1 in unroofed membranes and measured TNP-ATP binding (*Figure 4—figure supplement 1A*). We found only a small increase (approximately 1.5-fold) in apparent $EC_{50}$ compared to the same construct in the presence of SUR1 (37.6 µM and 25.6 µM respectively). Unfortunately, we were unable to achieve high enough expression of Kir6.2*-GFP alone to carry out PCF experiments in the absence of SUR1. However, we were able to measure currents from unlabelled Kir6.2-GFP alone (*Figure 4—figure supplement 1B*). As expected Kir6.2-GFP alone was much less sensitive to inhibition by TNP-ATP than Kir6.2-GFP + SUR1.

As Kir6.2*-GFP expression in the absence of SUR1 was not sufficient for PCF recordings, we took a mutational approach to better understand the role of SUR1 in inhibitory nucleotide binding. SUR1-K205 is located in the L0 linker of SUR1, which connects the first set of transmembrane domains (TMD0) to the ABC core structure (*Figure 1A*, *Figure 4A*; *Martin et al., 2017*; *Puljung, 2018*). This loop is adjacent to the inhibitory nucleotide-binding site on Kir6.2 and the interface between neighbouring Kir6.2 subunits. Mutations at SUR1-K205 were previously shown to reduce sensitivity of K_ATP to nucleotide-dependent inhibition (*Pratt et al., 2012*; *Ding et al., 2019*), and a recent cryo-EM stucture suggests that SUR1-K205 may directly coordinate the phosphates of ATP bound to Kir6.2

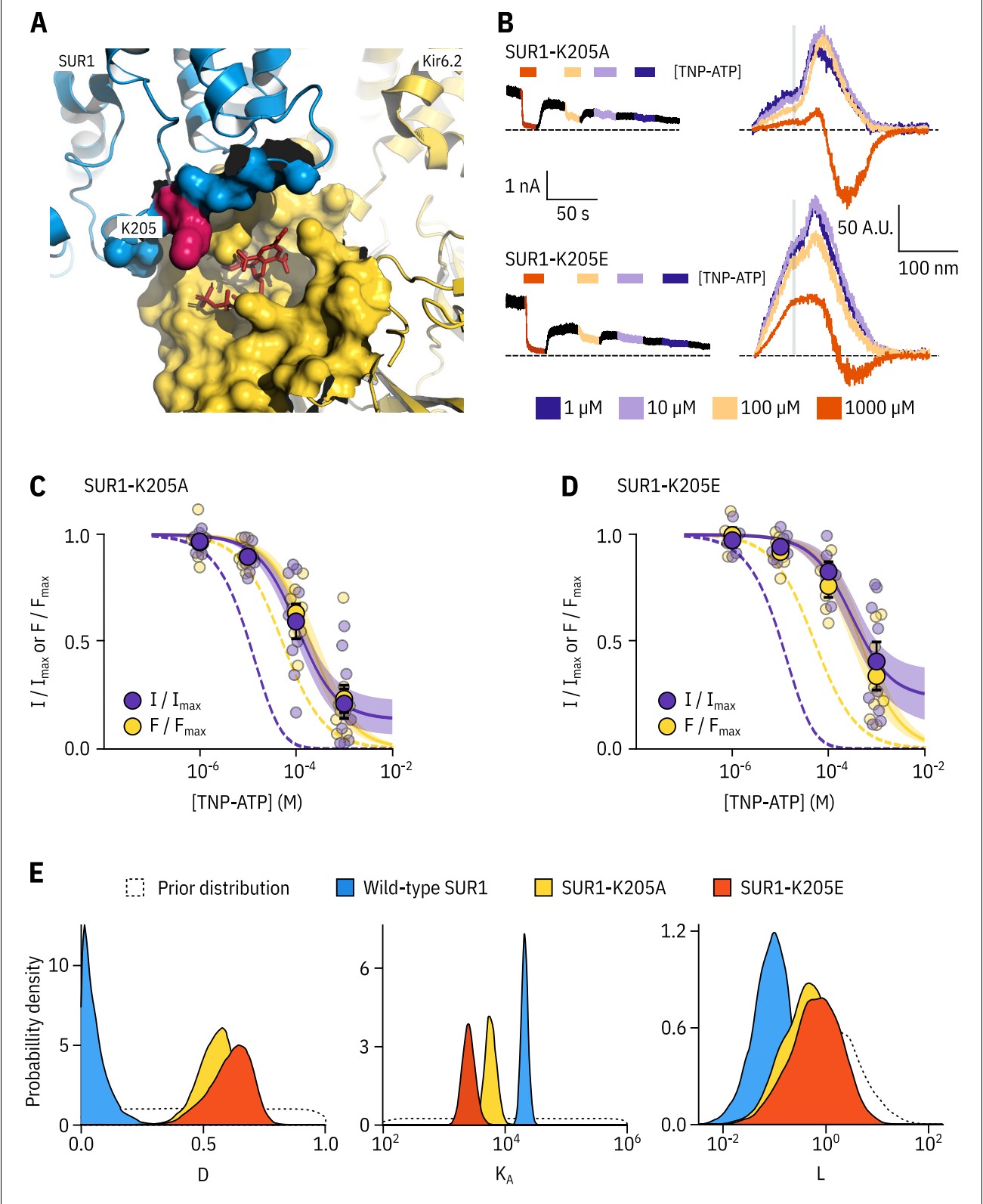

**Figure 4.** SUR1-K205 modulates both nucleotide affinity and inhibition of Kir6.2. (**A**) Hydrophobic surface representation of Kir6.2 (yellow, PDB accession #6BAA) and SUR1 (blue, PDB accession #6PZI). Residue K205 on SUR1 is highlighted in pink. As this residue was built as an alanine in the structure, we used the mutagenesis tool in PyMol to insert the native lysine residue. A docked TNP-ATP molecule is shown in red. (**B**) Representative current and fluorescence traces acquired simultaneously from excised patches expressing Kir6.2*-GFP with SUR1-K205A or SUR1-K205E. (**C,D**)

*Figure 4 continued on next page*

Figure 4 continued

Concentration-response for TNP-ATP inhibition of currents ($I/I_{max}$) and for quenching of ANAP fluorescence ($F/F_{max}$) in excised inside-out membrane patches expressing Kir6.2*-GFP + SUR1-K205A (C, n = 9) or Kir6.2*-GFP + SUR1-K205E (D, n = 9). Data were fit to the MWC-type model. Solid curves represent the median fits and shaded areas indicate the 95% quantile intervals. Fits to Kir6.2*-GFP + wild-type SUR1 are shown as dashed curves. (E) Posterior probability distributions for the full MWC-type model fit to Kir6.2*-GFP co-expressed with wild-type SUR1 (fits from *Figure 2*), SUR1-K205A and SUR1-K205E overlaid on the prior probability distribution.

The online version of this article includes the following figure supplement(s) for figure 4:

**Figure supplement 1.** SUR1 affects the apparent affinity for nucleotide binding to Kir6.2.

**Figure supplement 2.** Fixing the *L* parameter does not drastically affect the fits to the SUR1-K205A or SUR1-K205E data.

**Figure supplement 3.** Comparing the ability of each model to explain the data.

**Figure supplement 4.** A neutral choice of priors allows for the best fits to the data.

(*Ding et al., 2019*). Other mutations in L0 are associated with neonatal diabetes (*Ashcroft et al., 2017*) and PHHI (*Snider et al., 2013*).

We introduced a charge neutralization (alanine, K205A) and a charge reversal (glutamate, K205E) mutation at this position and measured simultaneous nucleotide binding and current inhibition with PCF (*Figure 4B,C,D*). The binding and inhibition curves for TNP-ATP almost perfectly overlaid for the SUR1-K205A mutant (*Figure 4C*). The same was also true for SUR1-K205E (*Figure 4D*). Data were fit with the MWC-type model as before. Mutating K205 to an alanine or a glutamate resulted in an apparent decrease in nucleotide binding affinity (*Figure 4C,D,E*). This was reflected by a decrease in the estimated $K_A$ for TNP-ATP, which correlated with the degree of conservation of the mutation, that is we observed a larger effect for the charge reversal compared to the charge neutralization mutation (*Figure 4E*). However, in addition to direct effects of K205 on nucleotide binding, we also observed a shift in *D* for both mutations (*Figure 4E*). This suggests a dual role for SUR1 in $K_{ATP}$ inhibition, both in contributing to nucleotide binding and in stabilising the nucleotide-bound closed state.

## Discussion

We have developed a novel approach that allows for site-specific measurement of nucleotide binding to $K_{ATP}$ and concomitant measurements of channel current. Performing these measurements simultaneously allowed us to examine nucleotide regulation of $K_{ATP}$ function in great detail. We used a Bayesian approach to fit models to our combined fluorescence/current data sets to extract meaningful functional parameters with a minimum of prior assumptions. Such insights would not be possible from experiments in which macroscopic currents or binding were measured in isolation.

PCF has been used successfully by other labs to simultaneously measure ligand binding and gating in HCN channels (*Biskup et al., 2007*; *Kusch et al., 2010*; *Wu et al., 2011*). These groups measured fluorescence from a cyclic nucleotide analogue that increased its quantum yield when bound, minimizing background fluorescence from unbound ligand. Additional background subtraction could be performed by imaging the patches using confocal microscopy such that a region corresponding to the patch membrane could be computationally selected, thus omitting background fluorescence from the surrounding solution (*Biskup et al., 2007*; *Kusch et al., 2010*). In our PCF experiments, we used a FRET-based approach to measure ligand binding. We acquired fluorescence emission spectra, such that donor fluorescence could be separated from acceptor fluorescence by wavelength. This allowed us to directly assess binding from the quenching of donor fluorescence, which was specific to $K_{ATP}$. FRET also provided the spatial sensitivity necessary to discriminate between nucleotide binding directly to Kir6.2 and to the nucleotide-binding sites of SUR1. We assume that any TNP-ATP bound non-specifically to our membranes would be too far from Kir6.2 to cause appreciable FRET. This assumption was confirmed by the lack of FRET between TNP-ATP and a Kir6.2*-GFP mutant (G334D), in which nucleotide binding was severely disrupted (*Figure 1H*).

Previous studies have suggested that $K_{ATP}$ inhibition follows an MWC-type model (*Trapp et al., 1998*; *Enkvetchakul and Nichols, 2003*; *Drain et al., 2004*; *Craig et al., 2008*; *Vedovato et al., 2015*). The majority of this earlier work was performed using single-channel measurements of mutated and/or concatenated channel subunits. In this study, we confirm these results using minimally perturbed channels with nucleotide sensitivity similar to that of wild-type $K_{ATP}$ (*Figure 1—*

*figure supplement 2A*). By using an MCMC approach to model fitting, we can also evaluate our models to assess how well the derived parameters were determined by the data. MCMC fits provide a basis for determining credible intervals for our parameter estimates. This allows for direct comparison of values derived from wild-type and different mutant constructs.

Although we did not explicitly include the effects of $PIP_2$ on $K_{ATP}$ gating in our model formulations, we assumed that the effects of $PIP_2$ on $P_{open}$ were implicitly modelled in our parameter $L$; rundown due to dissociation of $PIP_2$ manifests as a decrease in $L$ rather than a change in the number of channels. Although we were able to extract identifiable parameter estimates for $L$, $D$ and $K_A$, our estimates of $L$ for each model we considered were appreciably less well constrained than for the other parameters. We expect that this uncertainty arises from measuring a heterogeneous population of channels with regard to $PIP_2$ binding. Fixing $L$ to values obtained from the literature (*Figure 2—figure supplement 1*, *Figure 3—figure supplement 2*, *Figure 4—figure supplement 2*, *Figure 4—figure supplement 3*) allowed us to extract estimates for $D$ and $K_A$ that were functionally identical to those derived from unconstrained fits, suggesting that the uncertainty of $L$ does not affect our inferences for these other parameters. Although the value for $L$ that we obtained from the literature for Kir6.2*-GFP + SUR1 (0.8) is on the extreme end of the posterior probability distribution for $L$ from our fits, we would not expect it to change our estimates of $K_A$ or $D$ as these values were stable over a broad range of values of $L$ (*Figure 2—figure supplement 1*) for this construct. Therefore, PCF represents a robust means to compare $K_A$ and $D$ between different mutated $K_{ATP}$ constructs without worrying about the confounding effects of rundown.

Previous studies suggest that, whereas $K_{ATP}$ closure occurs via a concerted mechanism, individual nucleotide binding events at Kir6.2 are not equivalent (*Markworth et al., 2000*). Earlier attempts to determine the stoichiometry of inhibitory nucleotide binding to Kir6.2 (i.e. how many ATPs must bind to induce channel closure) have produced models ranging from those in which binding of a single nucleotide completely shuts $K_{ATP}$ to an MWC-type model in which each binding event is independent and contributes equally to channel closure (*Trapp et al., 1998*; *Markworth et al., 2000*; *Enkvetchakul and Nichols, 2003*; *Drain et al., 2004*; *Wang et al., 2007*; *Craig et al., 2008*; *Vedovato et al., 2015*). To resolve this controversy, we fit our data with both single-binding and MWC-type models. At very low values for $D$, such as we derived from our experiments, the predictions of both models are functionally very similar. Even in our MWC-type model, we expect most $K_{ATP}$ channels to be closed when just one molecule of nucleotide is bound.

It has been proposed that there is direct negative cooperativity between binding events at different subunits on Kir6.2 (*Wang et al., 2007*). We fit our data to an extended MWC-type model including an additional free parameter ($C$), representing negative binding cooperativity between subunits (*Figure 2—figure supplement 2*). Not surprisingly this model improved the fit to our data as assessed by the Bayes factor, which represents the marginal likelihood of one model over another to explain our observations (*Wagenmakers, 2007*; *Gronau et al., 2017*). We also tested the cooperative model using approximate leave-one-out cross validation, which assesses the ability of a model to predict new or out-of-sample data using in-sample fits (*Vehtari et al., 2017*). Although in this work, we are primarily concerned with the inferences made from our fits, the ability of a model to make predictions is a good measure of its usefulness. Based on this criterion, the cooperative model has no more predictive accuracy than either the MWC-type model or the single-binding model. Therefore, the inclusion of an additional free parameter is not justified. Furthermore, whereas the cooperative model yielded good fits with identifiable parameters for Kir6.2*-GFP + SUR1 channels, it failed to yield identifiable parameters for all the mutants considered. Thus, this model did not allow for direct comparison between constructs. However, it remains a possibility that these mutations function in part by abolishing binding cooperativity between subunits.

We performed all our experiments on mutated, tagged channels using a fluorescent derivative of ATP. This allowed us to fit mechanistic models and readily compare between mutated constructs that affect nucleotide inhibition of $K_{ATP}$. This raises an obvious question: how relevant are our findings to inhibition of wild-type $K_{ATP}$ by ATP? In a previous paper, we estimated $D$ and $K_A$ from an MWC-type model based on fits to published data for ATP inhibition of wild-type Kir6.2 + SUR1 (*Proks et al., 2010*; *Vedovato et al., 2015*). The value we obtained for $D$ (0.03) was quite similar to that we report here from our PCF measurements (0.04). We also obtained a similar estimate for $K_A$ in our previous model ($3.0 \times 10^4$ $M^{-1}$ vs $2.1 \times 10^4$ $M^{-1}$ from our PCF experiments). Despite obtaining similar parameters, past experiments in which only ionic currents were measured, did not allow us to

distinguish between competing gating models. Measuring currents and fluorescence simultaneously allowed for better model selection and aided in our ability to identify constrained parameters.

We compared the parameters derived for inhibitory nucleotide binding to those estimated for nucleotide activation of $K_{ATP}$ based on experiments in which currents and binding were measured in separate preparations (*Puljung et al., 2019*). In those experiments, we estimated a value for $E$, the factor by which binding of MgTNP-ADP to SUR1 stabilied channel opening, of 2.2. Although this value was derived using a different nucleotide, it still provides an approximate basis for comparing the coupling of nucleotide stimulation through SUR1 to nucleotide inhibition via binding to Kir6.2. If both activation and inhibition proceed via MWC-type models, the open closed equilibrium at saturating nucleotide concentrations is given by $L$ multiplied by $E^4$ or $D^4$, respectively. The degree of stabiliation of the open state of $K_{ATP}$ can be calculated as $-RT \ln E^4$ for activation. Stabiliation of the closed state is given by $-RT \ln D^4$. Based on our observations, saturating concentrations of MgTNP-ADP stabilied the open state by $-1.9$ kcal mol$^{-1}$ ($-7.9$ kJ mol$^{-1}$). At saturating concentrations, TNP-ATP stabilied the closed state of $K_{ATP}$ by $-7.9$ kcal mol$^{-1}$ (31.8 kJ mol$^{-1}$). Thus, assuming excitatory and inhibitory processes are independent, inhibition would be expected to dominate under conditions at which all the nucleotide binding sites are occupied. This is consistent with published measurements of wild-type $K_{ATP}$ in the presence of Mg$^{2+}$(*Proks et al., 2010*). In our previous study, we estimated $K_A$ for MgTNP-ADP binding to the stimulatory second nucleotide binding site of SUR1 to be $5.8 \times 10^4$ M$^{-1}$ ($K_D = 17\,\mu$M), higher affinity than the $K_A$ we report here for TNP-ATP binding to the inhibitory site on Kir6.2 ($2.1 \times 10^4$ M$^{-1}$, $K_D = 48\,\mu$M). Higher affinity binding to the stimulatory site may explain the ability of MgADP to increase $K_{ATP}$ currents in the presence of ATP (*Gribble et al., 1998*). This phenomenon may also explain the bell-shaped MgADP concentration-response curve for $K_{ATP}$, which shows an increase in current at low concentrations, followed by inhibition at higher concentrations (*Proks et al., 2010*; *Vedovato et al., 2015*). Future experiments in which activation and inhibition are measured by PCF for the same ligand will allow us to model the complex response of $K_{ATP}$ under conditions where all three nucleotide binding sites simultaneously affect channel gating (i.e. in the presence of Mg$^{2+}$).

Mutations that cause neonatal diabetes reduce the sensitivity of $K_{ATP}$ to nucleotide inhibition, and reduction in nucleotide sensitivity is broadly correlated with disease severity (*McTaggart et al., 2010*). We studied two residues on Kir6.2 that have been implicated in diabetes and have been proposed to affect nucleotide sensitivity via different mechanisms. We find that G334D drastically reduced the apparent affinity for nucleotide binding to $K_{ATP}$ in unroofed membranes. In our MWC-type models, this could only be explained by a dramatic decrease in $K_A$. This corroborates earlier hypotheses that mutating G334 directly disrupts inhibitory nucleotide binding to Kir6.2 (*Drain et al., 1998*). Due to poor expression, we were unable to test this construct using PCF. Therefore, we could not obtain accurate estimates of $K_A$ and $D$.

In contrast to G334D, the C166S mutation does not directly affect nucleotide binding to Kir6.2, but rather disrupts the ability of bound nucleotide to close the channel. This contributes to the decreased nucleotide sensitivity which was previously attributed solely to an increased $P_{open}$. In the future, we hope to use this rigorous approach to assess a whole panel of neonatal diabetes mutations in Kir6.2 to better understand the mechanism by which they cause disease.

Using PCF allowed us to probe more deeply into the role of SUR1 in regulating nucleotide inhibition of $K_{ATP}$. The cytoplasmic L0 loop of SUR1 was previously implicated in modulation of $P_{open}$ and nucleotide sensitivity of Kir6.2 (*Babenko and Bryan, 2003*; *Chan, 2003*; *Pratt et al., 2012*). We find that, in addition to directly contributing to tighter nucleotide binding at Kir6.2, SUR1 plays a critical role in preferentially stabilising the closed state of the channel when nucleotides are bound. Whereas a single nucleotide-binding event is sufficient for channel closure when Kir6.2 is associated with wild-type SUR1, mutating residue K205 reduced the ability of a single nucleotide to close the channel. This difference manifests in both our MWC-type and single-binding models (*Figure 4—figure supplement 3* and *Table 3*).

In addition to providing mechanistic insights into disease-associated mutations in Kir6.2, our PCF-based approach allows us to probe the interactions between Kir6.2 and SUR1 on two different levels. As we show here, we can use this method to examine the effects of SUR1 on inhibitory nucleotide binding to Kir6.2. We can also adapt this method to study activation of Kir6.2 by nucleotides bound to the stimulatory sites on SUR1. Mutations in SUR1 that cause neonatal diabetes may do so

by disrupting inhibitory binding/gating or enhancing the stimulatory effects of nucleotides. The formalism developed in this study provides a rigorous way to mechanistically assess the effects of these mutations. Our approach should be readily adaptable to the study of other nucleotide-gated channels including the cystic fibrosis transmembrane conductance regulator (CFTR, also an ABC-family protein) and purinergic P2X receptors.

# Materials and methods

## Key resources table

| Reagent type (species) or resource | Designation | Source or reference | Identifiers | Additional information |
|---|---|---|---|---|
| Cell line | HEK-293T (*H. sapiens*) | LGC Standards (ATCC CRL-3216) | | |
| Transfected construct (*Escherichia. coli*) | pANAP | Addgene | | |
| Transfected construct | pcDNA4/TO | Addgene | | |
| Transfected construct (*Aequorea victoria*) | pCGFP_EU | Gouaux Laboratory (Vollum Institute, Portland OR USA) | | |
| Transfected construct (*Homo sapiens*) | peRF1-E55D | Chin Laboratory (MRC Laboratory of Molecular Biology, Cambridge UK) | | |
| Antibody | Anti-HA High Affinity; Rat monoclonal antibody (clone 3F10) | Roche | (Roche Cat# 11867423001, RRID:AB_10094468) | (1:1000) |
| Antibody | Peroxidase-AffiniPure Goat Anti-Rat IgG (H + L) antibody | Jackson Immuno Research Labs | (Jackson ImmunoResearch Labs Cat# 112-035-003, RRID:AB_2338128) | Western blots: (1:20,000) Surface expression: (1:2000) |
| Chemical compound, drug | trinitrophenyl-ATP (TNP-ATP) | Jena Bioscience (Jena, Germany) | | |
| Chemical compound, drug | L-3-(6-acetylnaphthalen-2-ylamino)−2-aminopropionic acid | Asis Chemicals (Waltham, MA) | | |

## Molecular biology

Human Kir6.2 and SUR1 were subcloned into pcDNA4/TO and pCGFP_EU vectors for expression of wild-type and GFP-tagged constructs, respectively. pcDNA4/TO and pANAP were obtained from Addgene. peRF1-E55D and pCGFP_EU were kind gifts from the Chin Laboratory (MRC Laboratory of Molecular Biology, Cambridge, UK) and the Gouaux Laboratory (Vollum Institute, Oregon, USA) respectively. Amber stop codons and point mutations were introduced using the QuikChange XL system (Stratagene; San Diego, CA). All constructs were confirmed by DNA sequencing (DNA Sequencing and Services, University of Dundee, Scotland).

## Cell culture and channel expression

HEK-293T cells were obtained from and verified/tested for mycoplasma by LGC standards (ATTC CRL-3216, Middlesex, UK). Our working stock tested negative for mycoplasma contamination using the MycoAlert Mycoplasma Detection Kit (Lonza Bioscience; Burton on Trent, UK). Cells were plated onto either poly-L-lysine coated borosilicate glass coverslips (VWR International; Radnor, PA) or poly-D-lysine coated glass-bottomed FluoroDishes (FD35-PDL-100, World Precision Instruments). ANAP-tagged Kir6.2 constructs were labelled using amber stop codon suppression as described by *Chatterjee et al. (2013)*. Transfections were carried out 24 hr after plating using TransIT-LT1 (Mirus Bio LLC; Madison, WI) at a ratio of 3 µl per µg of DNA. Unless specified otherwise, all transfections included a Kir6.2 construct with an amber stop codon (TAG) at position 311 (Kir6.2-W311[TAG]), SUR1, pANAP and eRF1-E55D in the ratio 0.5:1.5:1:1. Transfected cells were cultured in Dulbecco's Modified Eagle Medium (Sigma; St. Louis, MO) + 10% foetal bovine serum, 100 U ml[-1] penicillin and

100 µg ml$^{-1}$ streptomycin (Thermo Fisher Scientific; Waltham, MA) supplemented with 20 µM ANAP (free acid, AsisChem; Waltham, MA). Cells were incubated at 33 ˚C and in the presence of 300 µM tolbutamide to enhance protein expression and channel trafficking to the plasma membrane (*Yan et al., 2007*; *Lin et al., 2015*). eRF1-E55D was included to increase efficiency of ANAP incorporation (*Schmied et al., 2014*). Experiments were carried out 2–4 days after transfection. We also expressed constructs labelled with ANAP at positions I182, F183, F198, and I210. Kir6.2-F183*, Kir6.2-F198*, and Kir6.2-I210* co-expressed with SUR1 did not produce sufficient currents for subsequent experimentation. Mutations at I182 are known to produce profound effects on nucleotide inhibition of K$_{ATP}$ (*Li et al., 2000*). Thus, we did not consider this site for further experimentation.

## Western blots

To confirm our ability to express full-length Kir6.2*-GFP, we performed western blots for HA-tagged Kir6.2 constructs in detergent-solubilized HEK-293T cells (*Figure 1—figure supplement 1C*). The HA tag plus a short linker (YAYMEKGITDLAYPYDVPDY) was inserted in the extracellular region following helix M1 of Kir6.2 between L100 and A101. Transfected HEK-293T cells grown in 6-well plates were harvested in cold PBS (Life Technologies Limited; Paisley, UK), pelleted at 0.2 x g for 2.5 min and resuspended in lysis buffer containing 0.5% Triton X-100, 100 mM potassium acetate, and a cOmplete protease inhibitor tablet (one tablet/50 ml, Roche; Basel, Switzerland), buffered to pH 7.4. After a 30 min benzonase (Sigma) treatment at room temperature, samples were mixed with a DTT containing reducing agent and loading buffer (NuPAGE, Invitrogen; Carlsbad, CA) and run on a precast Bis-Tris 4–12% poly-acrylamide gel at 200 V for 40 min. Proteins were wet transferred overnight onto polyvinylidene difluoride (PVDF) membranes (Immobilon P, Merck Millipore; Burlington, VT) in 25 mM Tris, 192 mM glycine, 20% methanol, and 0.1% SDS at 10 V on ice. Membranes were blocked with 5% milk in TBS-Tw (150 mM NaCl, 0.05% Tween 20, 25 mM Tris, pH 7.2) before staining for 30 min with a 1:1000 dilution of rat anti-HA monoclonal antibody in TBS-Tw (clone 3F10, Roche). After washing with TBS-Tw, membranes were incubated for 30 min with a 1:20,000 dilution of HRP-conjugated goat anti-rat polyclonal antibodies in TBS-Tw (Jackson ImmunoResearch; Ely, UK). Detection was performed using the SuperSignal West Pico Chemiluminescent Substrate (Thermo Fisher) and a C-DiGit Blot Scanner (Licor Biosciences; Lincoln, NE). Analysis was performed using custom code written in Python.

Transfection of wild-type Kir6.2-HA or Kir6.2-HA-GFP resulted in two bands on the western blots. The upper bands were close to the expected sizes for full-length Kir6.2-HA and Kir6.2-HA-GFP (46 kDa and 77kDa, respectively). We consistently observed a lower molecular weight band as well. This band must correspond to an N-terminally truncated Kir6.2 product, as the apparent molecular weight shifted with addition of the C-terminal GFP tag. Based on the molecular weight, we predict that the truncated protein product initiated from a start codon in the first transmembrane domain. Therefore, we believe it is unlikely that this protein would form functional channels or traffic to the plasma membrane. When Kir6.2-W311$^{TAG}$-HA or Kir6.2-W311$^{TAG}$-HA-GFP were co-transfected with SUR1, pANAP, and eRF1-E55D, and cells were cultured in the presence of ANAP, the western blots were similar to wild-type Kir6.2-HA or Kir6.2-HA-GFP. Over 90% full-length Kir6.2*-HA-GFP was produced under these conditions (*Figure 1—figure supplement 1D*). We were unable to quantify the percentage of full-length Kir6.2*-HA produced as the C-terminally truncated band resulting from termination at the TAG codon was very similar in size to the N-terminally truncated band. Co-expression with SUR1 increased the percentage of full-length Kir6.2*-HA-GFP produced (*Figure 1—figure supplement 1D*). In the absence of ANAP, we did not observe any full-length Kir6.2, indicating that there was no read-through of the amber (TAG) stop codon (*Figure 1—figure supplement 1D*).

## Confocal microscopy

Confocal imaging was performed using a spinning-disk system (Ultra-VIEW VoX, PerkinElmer; Waltham, MA) mounted on an IX81 microscope (Olympus; Southend-on-Sea, UK) with a Plan Apo 60x oil immersion objective (NA = 1.4), provided by the Micron Advanced Bioimaging Unit, Oxford. Transfected HEK-293T cells were incubated for 15 min with 1 nM CellMask Deep Red (Thermo Fisher) to stain plasma membranes before washing with PBS and imaging. ANAP was excited with a solid-state laser at 405 nm. GFP and CellMask were excited with an argon laser at 488 nm and 633 nm respectively. Images were captured on an EMCCD camera (ImagEM; Hamamatsu Photonics;

Welwyn Garden City, UK) binned at 2 × 2 pixels and analysed using Python. A median filter with a box size of 32 × 32 pixels was applied to improve the signal-to-noise ratio by reducing background fluorescence.

We examined the surface expression of our ANAP-labelled constructs using confocal microscopy (*Figure 1—figure supplement 1A,B*). When Kir6.2-W311$^{TAG}$-GFP was co-transfected with SUR1 along with pANAP and eRF1-E55D in the presence of ANAP, the ANAP and GFP fluorescence were co-localized at the plasma membrane. When wild-type Kir6.2-GFP was transfected under the same conditions, only GFP fluorescence was observed at the plasma membrane. ANAP fluorescence was diffuse and confined to the cytoplasm or intracellular structures. Thus, the plasma-membrane ANAP signal was specific for Kir6.2*-GFP.

## Surface expression assays

We measured surface expression of HA-tagged Kir6.2 subunits using an approach outlined by *Zerangue et al. (1999)* ( *Puljung et al., 2019*). Cells were plated on 19 mm coverslips coated with poly-L-lysine and transfected as described above. Following incubation, cells were rinsed with PBS before fixation with 10% formalin for 30 min at room temperature. After washing again, cells were blocked with 1% BSA in PBS for 30 min at 4 ˚C before a 1 hr incubation at 4 ˚C with a 1:1000 dilution (in PBS) of rat anti-HA monoclonal antibodies. Cells were then washed 5 times on ice with 1% BSA in PBS followed by a 30 min incubation at 4 ˚C with a 1:2000 dilution of HRP-conjugated goat anti-rat polyclonal antibodies. Cells were washed 5 times in PBS + 1% BSA and 4 times in PBS. Coverslips were removed from the culture dishes and placed in clean, untreated dishes for measurement. 300 µl of SuperSignal ELISA Femto Maximum Sensitivity Substrate (Thermo Fisher) was added to each sample and the luminescence was measured using a Glomax 20/20 Luminometer (Promega; Madison, WI) after a 10 s incubation.

HEK-293T cells were transfected with Kir6.2 constructs with or without a TAG stop codon corresponding to position 311. Cells were co-transfected with pANAP and eRF1-E55D in the presence or absence of SUR1 and cultured with or without ANAP. Wild-type Kir6.2-HA and Kir6.2-HA-GFP in the presence of SUR1 were included as positive controls. Kir6.2 constructs with no HA tag served as negative controls. In the presence of ANAP, we observed strong trafficking of Kir6.2*-HA-GFP to the plasma membrane, but much less trafficking of Kir6.2*-HA (*Figure 1—figure supplement 1E*). When cells were cultured in the absence of ANAP, we observed little to no Kir6.2 surface expression from cells that were transfected with Kir6.2-W311$^{TAG}$-HA or Kir6.2-W311$^{TAG}$-HA-GFP, suggesting that prematurely truncated constructs did not traffic to the plasma membrane. In the absence of SUR1, surface expression was weak for both wild-type and tagged constructs, despite the reported ability of Kir6.2-GFP to traffic to the plasma membrane in the absence of SUR1 (*John et al., 1998*; *Makhina and Nichols, 1998*).

## Epifluorescence imaging and spectroscopy

Epifluorescence imaging and spectroscopy were performed using a Nikon Eclipse TE2000-U microscope with a 60x water immersion objective (Plan Apo VC, NA = 1.2, Nikon; Kingston upon Thames, UK) or a 100x oil immersion objective (Nikon, Apo TIRF, NA = 1.49). Imaging of ANAP was performed using a 385 nm LED source (ThorLabs; Newton, NJ) with a 390/18 nm band-pass excitation filter, an MD416 dichroic and a 479/40 nm band-pass emission filter (all from ThorLabs). GFP was imaged using a 490 nm LED source (ThorLabs) with a 480/40 nm band-pass excitation filter, a DM505 dichroic, and a 510 nm long-pass emission filter (all from Chroma; Bellows Falls, VT). Fluorescence spectra were collected by exciting ANAP as above but using a 400 nm long-pass emission filter (ThorLabs), then passing emitted light through an IsoPlane 160 Spectrometer (Princeton Instruments; Trenton, NJ) with a 300 grooves mm$^{-1}$ grating. Images were collected with 1 s exposures on a Pixis 400BR_eXcelon CCD (Princeton Instruments).

## Electrophysiology

Patch pipettes were pulled from thick-walled borosilicate glass capillaries (GC150F-15, Harvard Apparatus; Holliston, MA) to a resistance of 1.5 MΩ to 2.5 MΩ when filled with pipette solution. Currents were recorded at −60 mV from excised inside-out patches using an Axopatch 200B amplifier equipped with a Digidata 1322A digitizer and using pClamp 10 software (Molecular Devices; San

Jose, CA). Currents were low-pass filtered at 5 kHz and digitized at 20 kHz. The bath solution (intra-cellular) contained 140 mM KCl, 10 mM HEPES, 1 mM EDTA and 1 mM EGTA (pH 7.3 with KOH). The pipette solution (extracellular) contained 140 mM KCl, 10 mM HEPES and 1 mM EDTA (pH 7.4 with KOH). All experiments were carried out in $Mg^{2+}$-free conditions. Currents were leak corrected using the current remaining in bath solution containing 5 mM barium acetate at 60 mV, assuming a linear leak with a reversal potential of 0 mV. Inhibition was calculated and corrected for rundown by alternating test concentrations of nucleotide solution with nucleotide-free solution, then expressing the test currents as a fraction of the average of the control currents before and after the test solution as described previously (*Proks et al., 2010*).

## FRET calculations

We calculated the expected FRET efficiency between ANAP incorporated at amino acid position 311 and a docked TNP-ATP molecule as described previously (*Puljung et al., 2019*). The equivalency between FRET efficiency (measured as ANAP quenching) and nucleotide binding is based on two main assumptions. Firstly, we assume that the observed quenching from a bound nucleotide does not differ dramatically between open and closed states of the channel. As there is no open-state structure of $K_{ATP}$, we do not know exactly how much relative movement would occur between a bound TNP-ATP and Kir6.2-W311. However, based on cryo-EM structures of apo and nucleotide-bound Kir6.2 we do not expect to see a change in the distance between these two positions (*Martin et al., 2019*).

Secondly, we assume that the ANAP and TNP-ATP molecules on each subunit do not undergo energy transfer with those on other subunits to an extent which would dramatically change the observed quenching. At saturating TNP-ATP concentrations, where each ANAP-labelled site on Kir6.2 is occupied, FRET between ANAP and the closest acceptor will be kinetically favoured and the overall FRET efficiency will not be affected by cross-talk between neighbouring sites (*Corry et al., 2005*). In the limiting case, at low TNP-ATP concentrations, one would expect a large proportion of Kir6.2 tetramers (with four ANAP-labelled binding sites) bound to only a single TNP-ATP molecule. In this case, we expect a 4% overestimation of nucleotide binding as calculated using a numerical method to simulate a single TNP-ATP acceptor with multiple ANAP donors based on the distances calculated from our docking (*Figure 1C*; *Deplazes et al., 2012*). This may have resulted in our binding curves becoming artifically shallow at low concentrations. However, this difference is not significant in the context of our measurements as it is smaller than the observed error of our measurements at low TNP-ATP concentrations.

## Unroofed binding measurements

Unroofed membranes were prepared as described previously (*Heuser, 2000*; *Zagotta et al., 2016*; *Puljung et al., 2019*). A coverslip plated with transfected HEK-293T cells was removed from the culture media and rinsed with PBS. The coverslip was then briefly sonicated using a probe sonicator (Vibra-cell; Newtown, CT) leaving behind adherent plasma membrane fragments. Cells cultured on FluoroDishes were rinsed and sonicated directly in the dish. Unroofed membrane fragments were nearly invisible in bright-field images and identified by their GFP and ANAP fluorescence. Fluorescent TNP-nucleotides (Jena Bioscience; Jena, Germany) were diluted in bath solution and perfused onto unroofed membranes using a valve- controlled microvolume superfusion system (µFlow, ALA Scientific Instruments; Farmingdale, NY).

Fluorescence spectra were collected as described above. A region of interest corresponding to the membrane fragment was manually selected and line-averaged for each wavelength. A similarly sized region of background was selected and averaged, then subtracted from the spectrum of interest. After subtraction, ANAP intensity was calculated by averaging the fluorescence intensity measured between 469.5 nm and 474.5 nm. Bleaching was corrected by fitting the normalised ANAP intensity of exposures taken during perfusion with nucleotide-free solution to a single exponential decay of the form

$$\frac{F}{F_{max}} = ae^{kt} + (1-a) \tag{1}$$

then using the fit to correct the intensity of exposures taken during perfusion with test nucleotide solutions.

Some experiments were excluded from further analysis due to obvious cross-contamination between different solutions within the μFlow superfusion system. These were identified by noticeable colour changes in the solution in the delivery tubes.

## Patch-clamp fluorometry

The tip of the patch pipette was centred on the slit of the spectrometer immediately after patch excision. Currents were measured as described above. Fluorescence emission spectra from the excised patch were acquired concurrently with current measurements, both during test solution application as well as nucleotide-free solution. Background subtraction was slightly imperfect due to the exclusion of TNP-ATP from the volume of the glass of the pipette, resulting in spectra that have negative intensities at the TNP-ATP peak at high nucleotide concentrations. However, this over-subtraction does not affect the size of the ANAP peak, which we used to quantify nucleotide binding.

ANAP bleaching was corrected as for the unroofed binding experiments with *Equation 1* (*Figure 2—figure supplement 3A*). Due to the lower signal-to-noise ratio for PCF compared to the unroofed membranes, we performed experiments from both high-to-low and low-to-high TNP-ATP concentrations to minimise artifacts from our bleaching corrections. Kir6.2*-GFP + SUR1 showed consistent bleaching time courses (*Figure 2—figure supplement 3B*) and an average of 34% of the initial ANAP fluorescence intensity remained at the end of each experiment (*Figure 2—figure supplement 3C*).

Some experiments were excluded from further analysis due to low fluorescence intensity, as we were concerned about a low signal to noise ratio influencing our results.

## Data processing and presentation

Raw spectrographic images and current traces were pre-processed in Python and Clampfit (Axon) before analysis with R (*R Development Core Team, 2019*). Where applicable, all experimental data points are displayed in each figure. The number of experiments is reported in the figure legends and tables. To help visualise uncertainty and prevent some data points being hidden, points are arranged with a small amount of horizontal jitter; vertical position remains unaffected. Unless otherwise stated, summary statistics are overlaid as the mean with error bars representing the standard error of the mean. Where these error bars are not visible, they are smaller than the size of the point used for the mean.

Hill fits to fluorescence quenching were nonlinear least-squares fits to the following equation:

$$\frac{y}{y_{max}} = E_{max} + \frac{1 - E_{max}}{1 + 10^{(EC_{50} - [TNPATP]) \cdot h}} \tag{2}$$

where $y$ represents normalised fluorescence intensity and $EC_{50}$ and $[TNPATP]$ are $\log_{10}$ values. Current inhibition data were fit to the same equation but with $y$ representing normalised current magnitude, $IC_{50}$ instead of $EC_{50}$, and $I_{max}$ instead of $E_{max}$.

## Bayesian model fitting

The MWC-type models considered (*Figure 2* and *Figure 2—figure supplement 2*) were formulated as follows:

$$\frac{F}{F_{max}} = \frac{K_A[TNPATP](1 + K_A[TNPATP])^3 + LDK_A[TNPATP](1 + DK_A[TNPATP])^3}{(1 + K_A[TNPATP])^4 + L(1 + DK_A[TNPATP])^4} \tag{3}$$

$$\frac{openchannels}{totalchannels} = \frac{L(1 + DK_A[TNPATP])^4}{(1 + K_A[TNPATP])^4 + L(1 + DK_A[TNPATP])^4} \tag{4}$$

When no ligand is present (i.e. when $[TNPATP] = 0$), *Equation 4* becomes:

$$\frac{openchannels}{totalchannels} = \frac{L}{1 + L} \tag{5}$$

We can use this to normalise the predicted changes in the open fraction to an observed change in current as:

$$\frac{I}{I_{max}} = \frac{L(1+DK_A[TNPATP])^4}{(1+K_A[TNPATP])^4 + L(1+DK_A[TNPATP])^4} \cdot \frac{1+L}{L} \tag{6}$$

Two variations on the full MWC model were also considered, and diagrammatic formulations are shown in *Figure 2—figure supplement 2*. The first was similar to the MWC-type model, except that the channels close after one molecule of TNP-ATP binding with subsequent binding events having no effect.

$$\frac{F}{F_{max}} = \frac{\begin{array}{c} LDK_A[TNPATP](1+3K_A[TNPATP]+3K_A^2[TNPATP]^2 \\ +K_A^3[TNPATP]^3) + K_A[TNPATP](1+K_A[TNPATP])^3 \end{array}}{\begin{array}{c} L(1+4DK_A[TNPATP]+6DK_A^2[TNPATP]^2+4DK_A^3[TNPATP]^3 \\ +DK_A^4[TNPATP]^4) + (1+K_A[TNPATP])^4 \end{array}} \tag{7}$$

$$\frac{I}{I_{max}} = \frac{\begin{array}{c} L(1+4DK_A[TNPATP]+6DK_A^2[TNPATP]^2+4DK_A^3[TNPATP]^3 \\ +DK_A^4[TNPATP]^4) \end{array}}{\begin{array}{c} L(1+4DK_A[TNPATP]+6DK_A^2[TNPATP]^2+4DK_A^3[TNPATP]^3 \\ +DK_A^4[TNPATP]^4) + (1+K_A[TNPATP])^4 \end{array}} \cdot \frac{1+L}{L} \tag{8}$$

The second alternate model was the same as the full MWC model, but with an additional term $C$ describing binding cooperativity between Kir6.2 subunits.

$$\frac{F}{F_{max}} = \frac{\begin{array}{c} LDK_A[TNPATP](1+3CDK_A[TNPATP]+3C^2D^2K_A^2[TNPATP]^2 \\ +C^3D^3K_A^3[TNPATP]^3) + K_A[TNPATP](1+3CK_A[TNPATP]) \\ +3C^2K_A^2[TNPATP]^2+C^3K_A^3[TNPATP]^3) \end{array}}{\begin{array}{c} L(1+4DK_A[TNPATP]+6CD^2K_A^2[TNPATP]^2+4C^2D^3K_A^3[TNPATP]^3 \\ +C^3D^4K_A^4[TNPATP]^4) + 1+4K_A[TNPATP]+6CK_A^2[TNPATP]^2 \\ +4C^2K_A^3[TNPATP]^3 + C^3K_A^4[TNPATP]^4 \end{array}} \tag{9}$$

$$\frac{I}{I_{max}} = \frac{\begin{array}{c} L(1+4DK_A[TNPATP]+6CD^2K_A^2[TNPATP]^2 \\ +4C^2D^3K_A^3[TNPATP]^3+C^3D^4K_A^4[TNPATP]^4) \end{array}}{\begin{array}{c} L(1+4DK_A[TNPATP]+6CD^2K_A^2[TNPATP]^2 \\ +4C^2D^3K_A^3[TNPATP]^3+C^3D^4K_A^4[TNPATP]^4) \\ +1+4K_A[TNPATP]+6C^2K_A^2[TNPATP]^2 \\ +4C^2K_A^3[TNPATP]^3+C^3K_A^4[TNPATP]^4 \end{array}} \cdot \frac{1+L}{L} \tag{10}$$

Each model was fit to the combined patch-clamp fluorometry datasets using the brms package (*Gelman et al., 2015*; *Burkner, 2017*) in R. Prior probability distributions for each parameter were supplied as:

$$\begin{array}{ll} \log_{10}(L) & \sim Normal(\mu:0, \sigma^2:0.7) \\ D & \sim Uniform(min:0, max:1) \\ \log_{10}(K_A) & \sim Uniform(min:2, max:6) \\ C & \sim Uniform(min:0, max:1) \end{array} \tag{11}$$

so that all priors are flat apart from L, which is weakly informative with 99% of its density falling between unliganded open probabilities of 0.01 and 0.99, and 85% falling between 0.1 and 0.9.

We considered two alternative sets of priors for fitting the MWC-type model to our mutant constructs (*Figure 4—figure supplement 4*). We generated a narrow informative prior by fitting normal distributions to the posterior probability density of our fits to Kir6.2*-GFP + SUR1, and a broad informative prior by increasing the standard deviation of the fitted normal distribution by a factor of ten (*Figure 4—figure supplement 4A*). Using narrow informative priors results in poorer fits as it does

not allow for high enough values of $D$ to explain the data, whereas the broad informative priors result in fits which do not visibly differ much from the neutral priors (*Figure 4—figure supplement 4B*). Whereas the fits using the broad informative priors were visually similar to those using neutral priors, the posterior probability distributions for the parameters were slightly different (*Figure 4—figure supplement 4C*). Notably, due to the broad prior distribution supplied for $L$, the posterior probability distribution for $L$ is also very broad for each construct, with a large amount of probability density over unlikely values, that is over half the probability density for Kir6.2*-GFP + SUR1-K205A and Kir6.2*-GFP + SUR1-K205E was for $L$ values lower than 0.01 (corresponding to a biologically implausible unliganded $P_{open}$ <0.01).

Each model was run with four independent chains for 10,000 iterations each after a burn-in period of 20,000 iterations, saving every 10th sample for a total of 4000 samples per model. Each model parameter achieved a minimum effective sample size of 3500 and a potential scale reduction statistic ($\hat{R}$) of 1.00. Where applicable, the posterior probabilities of each parameter are reported as the median and the 95% equal-tailed interval. Bayes factors were calculated using bridge-sampling (*Gronau et al., 2017*), and leave-one-out cross-validation (LOO-CV) was performed using the loo package (*Vehtari et al., 2017*).

### Docking

Computational docking of TNP-ATP into the nucleotide binding site of Kir6.2 was performed using AutoDock-Vina (*Trott and Olson, 2010*) and Pymol (Schrödinger, LLC; New York, NY). 11 TNP-ATP structures from the Protein Data Bank (PDB accession #s 1I5D, 3AR7, 5NCQ, 5SVQ, 5XW6, 2GVD, 5A3S, 2PMK, and 3B5J) were used as starting poses and a 15 Å × 11.25 Å × 15 Å box was centred on the ATP bound to Kir6.2 in PDB accession #6BAA (*Martin et al., 2017*). Protonation states for each residue were assigned using PDB2PQR and PROPKA 3.0 (*Dolinsky et al., 2004*). The modal highest-scoring pose from the docking run was selected (PDB accession #5XW6, *Kasuya et al., 2017*) and distances were measured from a pseudo atom at the centre of the fluorescent moiety. TNP-ATP (PDB #3AR7, *Toyoshima et al., 2011*) was positioned into the first nucleotide binding domain of SUR1 (PDB #6PZI, *Martin et al., 2019*) using the alignment tool in Pymol.

### Chemicals and stock solutions

Unless otherwise noted, all chemicals were obtained from Sigma. TNP-ATP was obtained as a 10 mM aqueous stock from Jena Bioscience and stored at −20˚ C aqueous stocks of ANAP-TFA were prepared by dissolving the free acid in 30 mM NaOH, and were stored at −20˚ C. Tolbutamide stocks (50 mM) were prepared in 100 mM KOH and stored at −20˚ C.

## Acknowledgements

We wish to thank Raul Terron Exposito for technical assistance and Dr. Natascia Vedovato for helpful discussions. Dr. James Cantley provided access to the Licor scanner for western blots. This work was supported by the Biotechnology and Biological Science Research Council (BB/R002517/1) and the Wellcome Trust Oxion graduate program.

## Additional information

### Funding

| Funder | Grant reference number | Author |
|---|---|---|
| Biotechnology and Biological Sciences Research Council | BB/R002517/1 | Frances M Ashcroft Michael C Puljung |
| Wellcome | 203731/Z/16/A | Samuel G Usher |

The funders had no role in study design, data collection and interpretation, or the decision to submit the work for publication.

## Author contributions
Samuel G Usher, Conceptualization, Data curation, Software, Formal analysis, Investigation, Visualization, Methodology; Frances M Ashcroft, Conceptualization, Resources, Supervision, Funding acquisition, Project administration; Michael C Puljung, Conceptualization, Resources, Software, Formal analysis, Supervision, Funding acquisition, Methodology, Project administration

## Author ORCIDs
Samuel G Usher Ⓜ https://orcid.org/0000-0002-2487-6547
Frances M Ashcroft Ⓜ https://orcid.org/0000-0002-6970-1767
Michael C Puljung Ⓜ https://orcid.org/0000-0002-9335-0936

## Decision letter and Author response
Decision letter https://doi.org/10.7554/eLife.52775.sa1
Author response https://doi.org/10.7554/eLife.52775.sa2

# Additional files

## Supplementary files
• Transparent reporting form

## Data availability
All data and associated code are available on github (https://github.com/smusher/KATP_paper_2019; copy archived at https://github.com/elifesciences-publications/KATP_paper_2019) and have also been uploaded to Dryad (https://doi.org/10.5061/dryad.0vt4b8gtv).

The following dataset was generated:

| Author(s) | Year | Dataset title | Dataset URL | Database and Identifier |
|---|---|---|---|---|
| Usher SG, Ashcroft FM, Puljung MC | 2019 | Nucleotide inhibition of the pancreatic ATP-sensitive K+ channel explored with patch-clamp fluorometry | https://doi.org/10.5061/dryad.0vt4b8gtv | Dryad Digital Repository, 10.5061/dryad.0vt4b8gtv |

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
