## [Decision Letter]

**Acceptance summary:**

This paper describes experiments to investigate the mechanism of nucleotide inhibition of K_ATP_ channels important for insulin secretion from the pancreas and neonatal diabetes. K_ATP_ channels are heteromers of KIR6.2 pore-forming subunits and SUR1 modulatory subunits, and they contain 3 different classes of ATP binding sites, two excitatory sites on SUR1 and one inhibitory site on KIR6.2. This study uses a set of powerful methods including noncanonical amino acid incorporation, fluorescence resonance energy transfer (FRET), and patch-clamp fluorometry (PCF) to simultaneously measure the binding of a fluorescent derivative of ATP (TNP-ATP) specifically to the inhibitory site in KIR6.2, and the inhibition of the ionic current. Somewhat surprisingly, the authors found that the inhibition of the current occurs with an affinity about an order of magnitude higher than the binding affinity of ATP to the inhibitory site. This seemingly paradoxical result is explained using a gating model based on the allosteric mechanism of Monod-Wyman-Changeux (MWC) for protein allostery where the binding of just the first ATP almost completely inhibits the channel. They then go on to use their experimental and analysis methods to determine the effects of two different childhood diabetes mutations and the effect of the SUR1 subunit on gating.

**Decision letter after peer review:**

Thank you for submitting your article "Nucleotide inhibition of the pancreatic ATP-sensitive K^+^ channel explored with patch-clamp fluorometry" for consideration by *eLife*. Your article has been reviewed by three peer reviewers, and the evaluation has been overseen by a Reviewing Editor and Richard Aldrich as the Senior Editor. The reviewers have opted to remain anonymous.

The reviewers have discussed the reviews with one another and the Reviewing Editor has drafted this decision to help you prepare a revised submission.

Summary:

This manuscript describes experiments to investigate the mechanism of nucleotide inhibition of K_ATP_ channels important for insulin secretion from the pancreas and neonatal diabetes. K_ATP_ channels are heteromers of KIR6.2 pore forming subunits and SUR1 modulatory subunits and contain 3 different classes of ATP binding sites, two excitatory sites on SUR1 and one inhibitory site on KIR6.2. This study uses a set of powerful methods including noncanonical amino acid incorporation, fluorescence resonance energy transfer (FRET), and patch-clamp fluorometry (PCF) to simultaneously measure the binding of a fluorescent derivative of ATP (TNP-ATP) specifically to the inhibitory site in KIR6.2, and the inhibition of the ionic current. These authors found, somewhat surprisingly, that the inhibition of the current occurs with an affinity about an order of magnitude higher than the binding affinity of ATP to the inhibitory site. This seemingly paradoxical result is explained using a gating model based on the allosteric mechanism of Monod-Wyman-Changeux (MWC) for protein allostery where the binding of just the first ATP almost completely inhibits the channel. They then go on to use their experimental and analysis methods to determine the effects of two different childhood diabetes mutations and the effect of the SUR1 subunit on gating.

The PCF data are beautiful and the paper is very well-written.

Essential revisions:

1) Please address the discrepancy between your conclusion that inhibition by TNP-ATP is weak and incomplete for the C166S mutation, and previous reports that (unmodified) ATP at 10 mM produces complete inhibition of this mutant (for instance, in pmid 11159439 and other papers). Your raw data in Figure 3D are compatible with strong inhibition at high concentration, but your model results are not (Figure 3D/E). It seems important that you address this, ideally with some new data at higher [TNP-ATP], but at the very least by discussion and perhaps additional modeling to learn how more complete inhibition would affect the conclusions.

Related to this, please explain the basis for selecting the prior distribution of the parameters for the modeling. Might it make sense to include information about the WT parameters in the prior distribution when fitting the mutant data?

2) The kinetics work is the weakest part of the paper. There are no controls for the minimum time required to exchange the solution by perfusion, there is barely one time point to constrain the on rate at 100 µM, and the assignment of the rates are model dependent. We think that this section should be removed.

---

## [Author Response]

Essential revisions:1) Please address the discrepancy between your conclusion that inhibition by TNP-ATP is weak and incomplete for the C166S mutation, and previous reports that (unmodified) ATP at 10 mM produces complete inhibition of this mutant (for instance, in pmid 11159439 and other papers). Your raw data in Figure 3D are compatible with strong inhibition at high concentration, but your model results are not (Figure 3D/E). It seems important that you address this, ideally with some new data at higher [TNP-ATP], but at the very least by discussion and perhaps additional modeling to learn how more complete inhibition would affect the conclusions.

There have been numerous previous studies examining the effect of amino acid substitutions at Kir6.2-C166 on the sensitivity of K_ATP_ to nucleotide inhibition. Some report complete inhibition at high ATP concentrations (Kir6.2,N160D,C166S + SUR1, Enkvetchakul et al., 2000) whereas others only report partial inhibition (Kir6.2,C166S-△C26, Kir6.2,C166S + SUR1, Trapp et al., 1998; Kir6.2,C166A-GFP + SUR1, Ribalet et al., 2006). The maximal block at high nucleotide concentrations appears to be background dependent.

To directly address this issue in our construct, Kir6.2*,C166S-GFP + SUR1, we applied 10 mM ATP to excised patches and observed only partial inhibition. We have added these data to Figure 3D. We did not measure inhibition by TNP-ATP at concentrations > 1 mM as our TNP-ATP was purchased as a triethylammonium salt (of indeterminate molar ratio of TNP-ATP:triethylammonium). We found that triethylammonium inhibits both Kir6.2*-GFP +SUR1 and Kir6.2*,C166S-GFP + SUR1 at millimolar concentrations. Thus, we would expect it to add to the total amount of current inhibition at high TNP-ATP concentrations. We have added a new supplement (Figure 3—figure supplement 2A,B) to demonstrate this.

We also performed additional modelling (Figure 3—figure supplement 2C,D) to illustrate that at relatively high values for *L*D^4^* the MWC-type model predicts a nucleotide-insensitive plateau of current at higher concentrations rather than a *bona fide* right shift in the inhibition curve relative to the binding curve. The height of the plateau at saturating nucleotide concentrations is proportional to the unliganded open probability of the channel, which may help to explain the diversity of results from the existing literature. We have included additional discussion in the text (paragraph four subsection “Kir6.2-C166S affects the ability of bound nucleotides to close K_ATP_.”).

Related to this, please explain the basis for selecting the prior distribution of the parameters for the modeling. Might it make sense to include information about the WT parameters in the prior distribution when fitting the mutant data?

We have now repeated our fits to the mutant data using two additional sets of priors. We used Gaussian fits to the posterior probabilities from the wild-type data as our first set. This resulted in parameter estimates that were very similar to wild-type, but the resulting curves fit the data poorly. We generated an additional set of priors using the centres of the Gaussian fits to the wild-type posterior probabilities, but with a 10-fold larger standard deviation (i.e. broader distributions centred on the same values). The resulting fits were similar to those generated from our relatively unbiased priors, with the exception of our estimates for *L*, which gave too much weight to unrealistic values for unliganded open probability (<1%). Thus, we believe we have chosen our priors well. We have included these new fits as Figure 4—figure supplement 4 and added additional discussion in the text (subsection “Bayesian model fitting”).

2) The kinetics work is the weakest part of the paper. There are no controls for the minimum time required to exchange the solution by perfusion, there is barely one time point to constrain the on rate at 100 uM, and the assignment of the rates are model dependent. We think that this section should be removed.

We agree with your assessment and have removed this section from the paper and edited the Materials and methods accordingly (subsection “Epifluorescence imaging and spectroscopy.”). We moved the remaining panels from the original Figure 4 to Figure 4—figure supplement 1, and renumbered other figures accordingly.